# Hybridizing anomalous Nernst effect in artificially tilted multilayer based on magnetic topological material

Takamasa Hirai [1] ✉, Fuyuki Ando [1], Hossein Sepehri-Amin [1] & Ken-ichi Uchida [1,2] ✉

Transverse thermoelectric conversion holds significant potential in addressing complex challenges faced by classical Seebeck/Peltier modules. A promising transverse thermoelectric phenomenon is the anomalous Nernst effect originating from nontrivial band structures in magnetic topological materials. However, the currently reported performance of the anomalous Nernst effect in topological materials, e.g., $Co_2MnGa$, remains insufficient for practical thermoelectric applications. Here, we unveil an unconventional availability of the anomalous Nernst effect by integrating magnetic topological materials into artificially tilted multilayers, known to exhibit the structure-induced transverse thermoelectric conversion due to the off-diagonal Seebeck effect. Our experiments reveal that the transverse thermoelectric performance in $Co_2MnGa$-based artificially tilted multilayers is improved through the hybrid action of the anomalous Nernst and off-diagonal Seebeck effects, with the magnetization-dependent performance modulation being one order of magnitude greater than the performance achievable with the anomalous Nernst effect alone. This synergy underscores the importance of hybrid transverse thermoelectric conversion and paves a way for advancing thermoelectric applications using magnetic materials.

Developments in thermoelectric technologies have garnered attention for their capability to directly convert a heat current $\mathbf{J}_q$ into a charge current $\mathbf{J}_c$ and vice versa, enabling power generation from wasted/environmental heat and solid-state refrigeration without mechanical moving parts[1–4]. Conventional thermoelectric devices are based on the Seebeck effect and its Onsager reciprocal, that is, the Peltier effect, in which $\mathbf{J}_c$ and $\mathbf{J}_q$ flow in the same direction. Due to this longitudinal geometry, a thermoelectric module based on the Seebeck/Peltier effect usually consists of a bunch of $p$-type and $n$-type conductors connected electrically in series and thermally in parallel with many electrodes to enlarge the thermoelectric output. Although the Seebeck/Peltier module is progressing to the daily market, its complex three-dimensional structure remains indispensable problems, e.g., low mechanical endurance and energy loss due to many

electrode junctions and their contact electrical and thermal resistances.

Transverse thermoelectric conversion has practical advantages over longitudinal thermoelectric effects[4]. The transverse thermoelectric effects allow interconversion between $\mathbf{J}_c$ and $\mathbf{J}_q$ in the orthogonal direction. Separating the $\mathbf{J}_c$ and $\mathbf{J}_q$ directions enables the enhancement of thermoelectric output simply by upscaling the size of a single thermoelectric material, which can reduce the number of electrodes and junctions in modules[4]. The ordinary Nernst effect (ONE) is a representative transverse thermoelectric effect; it is usually obtained in nonmagnetic conductors under an external magnetic field $\mathbf{H}$, referring to the generation of $\mathbf{J}_c$ in the cross-product direction of the applied $\mathbf{J}_q$ and $\mathbf{H}$ owing to the Lorentz force. The ordinary Ettingshausen effect (OEE) is the Onsager reciprocal of ONE. In magnetic

[1]National Institute for Materials Science, Tsukuba 305-0047, Japan. [2]Department of Advanced Materials Science, Graduate School of Frontier Sciences, The University of Tokyo, Kashiwa 277-8561, Japan. ✉e-mail: HIRAI.Takamasa@nims.go.jp; UCHIDA.Kenichi@nims.go.jp

materials, the anomalous Nernst (Ettingshausen) effect manifests via the momentum-space Berry curvature related to the spin-orbit interaction and/or spin-dependent scattering acting on conduction carriers, where $J_c$ ($J_q$) is generated in the cross-product direction of the applied $J_q$ ($J_c$) and a spontaneous magnetization $M$, which is referred to as ANE (AEE)[5,6]. Unlike ONE/OEE, which requires a substantial magnitude of $H$ to generate a large transverse thermopower, ANE/AEE offers an advantage of operating at much smaller magnetic field values or in the absence of $H$ when magnetic materials have finite remanent magnetization. Such features and functionalities provided by ANE/AEE have motivated advanced research on spin caloritronics and condensed matter physics[7-14]. While ONE has been shown to scale with carrier mobility[15], such simple scaling rules do not exist for ANE, and the material exploration for ANE is underway from various perspectives. By spotlighting the intrinsic Berry curvature contribution, several promising magnetic topological materials for transverse thermoelectrics have been discovered[16-25]. However, the thermopower due to ANE is still < 8 µV/K and the record-high dimensionless figure of merit $zT$ for ANE is approximately $7 \times 10^{-4}$ at the temperature $T \sim 300$ K[26], which is several orders of magnitude smaller than that for the Seebeck effect.

In this study, we provide a novel practical possibility for ANE/AEE in transverse thermoelectrics by hybridizing it into artificially tilted multilayers (ATMLs). The transverse thermoelectric conversion is not a unique feature of the Nernst/Ettingshausen effects but appears in anisotropic materials without requiring $H$ or $M$. ATML is one example of such anisotropic materials and consists of alternately and obliquely stacked two different conductors[27-36]. The geometrically tilted structure generates finite off-diagonal components in the thermoelectric transport tensors owing to anisotropic carrier flows; thus, it is usually called the off-diagonal Seebeck/Peltier effect (ODSE/ODPE). Here, the sign of the thermoelectric output due to ODSE/ODPE can be reversed by the 180° reversal of the relative angle between the tilting direction and input heat/charge current. Because the origin of ODSE/ODPE is the Seebeck/Peltier coefficients of constituent materials, a large transverse conversion by ODSE/ODPE has been realized using conventional thermoelectric materials[27-36]. Herein, we report hybrid transverse magneto-thermoelectric conversion by superimposing ANE/AEE on ODSE/ODPE in ATMLs consisting of magnetic topological material/thermoelectric material stacks (Fig. 1). Although the hybrid transverse magneto-thermoelectric conversion based on ONE/OEE and ODSE/ODPE in ATMLs comprising nonmagnetic thermoelectric materials, i.e., Bi-Sb and Bi-Sb-Te, has recently been reported[37], hybridizing ANE/AEE and its feature superior to ONE/OEE has not been demonstrated and magnetic topological materials have not been incorporated into

AMTLs so far. By means of the thermoelectric imaging technique based on lock-in thermography (LIT), we visualize the spatial distribution of transverse thermoelectric charge-to-heat current conversion processes in ATMLs based on a magnetic topological material and obtain large transverse thermoelectric cooling. Systematic LIT experiments separate the contribution of $M$-dependent AEE from $M$-independent ODPE, confirming the $M$-induced tuning of transverse thermoelectric conversion by hybridizing ANE/AEE in the magnetic topological material. Thermopower measurements reveal that the ANE-induced modulation of $zT$ for transverse thermoelectric conversion in ATMLs is several times larger than $zT$ for ANE alone in single magnetic materials ever reported, owing to the existence of $M$-independent ODSE. The development of ATML-based transverse magneto-thermoelectrics will boost research on thermoelectric device applications as well as topological materials science.

## Results

### Fabrication of Co$_2$MnGa-based artificially tilted multilayers

As the magnetic component of ATML, we selected a Weyl ferromagnet $Co_2MnGa$ alloy, which is one of the magnetic topological materials and exhibits not only a top-level transverse thermopower due to ANE (6-8 µV/K) owing to its topological feature but also a relatively large negative Seebeck coefficient $S_S$ (around −30 µV/K) among ferromagnetic conductors[19,26,38-40]. The thermoelectric component of ATML was chosen from a viewpoint of improving ODSE/ODPE in ATML. Generally, a combination of two materials with large difference in $S_S$ and with significantly different electrical and thermal conductivities ($\sigma$ and $\kappa$, respectively) enhances the transverse thermoelectric performance for ODSE/ODPE in ATML[29-37]. Thus, as the thermoelectric component of $Co_2MnGa$-based ATML, we mainly focused on $p$-type $Bi_{0.2}Sb_{1.8}Te_3$, which exhibits large positive $S_S$ ( > 100 µV/K) and much smaller $\sigma$ and $\kappa$ than $Co_2MnGa$[41].

To determine the structure of ATML, we analytically calculated the transverse thermoelectric conversion performance of $Co_2MnGa$/$Bi_{0.2}Sb_{1.8}Te_3$ ATML through measuring the electrical, thermal, and thermoelectric transport properties of single $Co_2MnGa$ and $Bi_{0.2}Sb_{1.8}Te_3$ alloys (see Methods). The transport properties of our polycrystalline $Co_2MnGa$ and $Bi_{0.2}Sb_{1.8}Te_3$ alloys prepared by spark plasma sintering (SPS) are summarized in Supplementary Table 1. The $\sigma$ and $S_S$ values at a magnetic field of 0 T and 0.8 T are almost identical in $Co_2MnGa$ and $Bi_{0.2}Sb_{1.8}Te_3$, indicating that the ordinary/anisotropic magnetoresistance and magneto-Seebeck effects are negligibly small in these materials below 0.8 T. The $\kappa$ value of $Co_2MnGa$ is much larger than that of $Bi_{0.2}Sb_{1.8}Te_3$; such a large difference in $\kappa$ makes it suitable for constructing ATMLs. Using $\sigma$, $\kappa$, and $S_S$ values of the $Co_2MnGa$ and $Bi_{0.2}Sb_{1.8}Te_3$ slabs, the electrical and thermal conductivities orthogonal to each other ($\sigma_{xx}$ and $\kappa_{yy}$) and the transverse thermopower due to ODSE ($S_{OD}$) were simulated for $Co_2MnGa$/$Bi_{0.2}Sb_{1.8}Te_3$ ATML, according to the equations in ref. 36 (see Supplementary Note 1). Figure 2a-d respectively display the contour maps for $S_{OD}$, $\sigma_{xx}$, $\kappa_{yy}$, and $zT$ for simulated ODSE, $z_{OD}T$ ($= S_{OD}^2 \sigma_{xx} T/\kappa_{yy}$), in $Co_2MnGa$/$Bi_{0.2}Sb_{1.8}Te_3$ ATML as functions of a tile angle $\theta$ and thickness ratio $R = t_M/(t_{TE} + t_M)$ at the zero magnetic field, where $t_{M(TE)}$ is the thickness of magnetic (thermoelectric) layer and $T = 300$ K was used. Here, it is found that the values of $\theta \sim 30°$ and $R \sim 0.5$ in $Co_2MnGa$/$Bi_{0.2}Sb_{1.8}Te_3$ ATML satisfy the optimal $z_{OD}T$ (~0.13), which is much larger than $zT$ for ANE in single-magnetic materials.

Based on the above simulation, $Co_2MnGa$-based ATML was prepared by following procedures (see also Methods). Initially, we prepared a cylindrical $Co_2MnGa$ slab with a diameter of 20 mm through the SPS method, subsequently slicing it into several disks each with $t_M$ of 0.75 mm. Then, these sliced $Co_2MnGa$ discs and Bi-Sb-Te powders were alternately piled up and bonded together via SPS. Notably, the Bi-Sb-Te powder used here is identical to that utilized in sintering the single $Bi_{0.2}Sb_{1.8}Te_3$ slab. Here, we confirmed that the sharp repeating

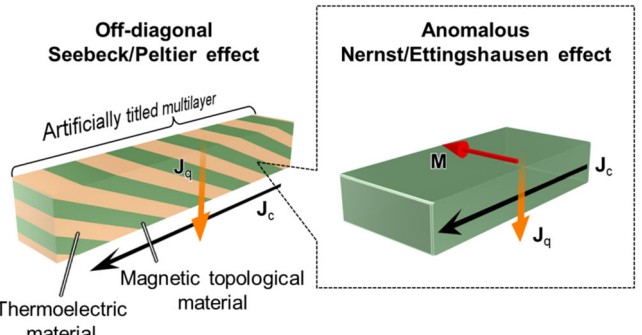

**Fig. 1 | Hybrid transverse magneto-thermoelectric conversion in artificially tilted multilayers using magnetic topological materials.** Schematics of the off-diagonal Seebeck/Peltier effect (ODSE/ODPE) in an artificially tilted multilayer (ATML) comprising magnetic and thermoelectric materials and the anomalous Nernst/Ettingshausen effect (ANE/AEE) in the magnetic material. Here, $J_c$, $J_q$, and $M$ denote the charge current, heat current, and magnetization, respectively.

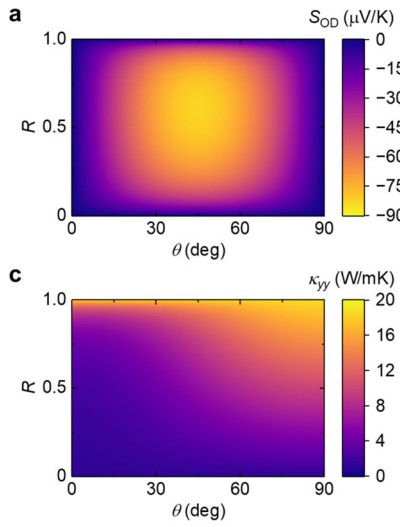

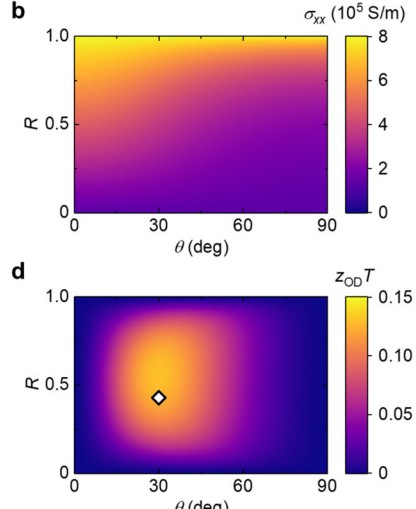

**Fig. 2 | Simulation of transverse thermopower and electrical/thermal conductivity for Co₂MnGa/Bi₀.₂Sb₁.₈Te₃ ATML. a–d** Contour maps depicting the transverse thermopower due to ODSE $S_{OD}$ (**a**), electrical conductivity along the $\mathbf{J_c}$ direction $\sigma_{xx}$ (**b**), thermal conductivity along the $\mathbf{J_q}$ direction $\kappa_{yy}$ (**c**), and dimensionless figure of merit for ODSE $z_{OD}T$ (= $S_{OD}^2 \sigma_{xx} T / \kappa_{yy}$) at zero magnetic field and

the temperature $T$ = 300 K (**d**) as functions of the tilt angle of ATML $\theta$ and thickness ratio $R$ defined as $R = t_M/(t_M + t_{TE})$ with $t_{M(TE)}$ being the thickness of the magnetic (thermoelectric) layer. The open diamond symbol in (**d**) represents the $\theta$ (= 30° ± 1°) and $R$ (= 0.43) values for our sample. $\theta$ was estimated from the thermal images obtained for lock-in thermography (LIT).

boundaries in the Co₂MnGa/Bi₀.₂Sb₁.₈Te₃ stack through structural characterizations; the composition of the bulk regions of each layer was uniform and the atomic interdiffusion was limited to a region of ~10 µm at the Co₂MnGa/Bi₀.₂Sb₁.₈Te₃ interfaces (Supplementary Fig. 1). Finally, the Co₂MnGa/Bi₀.₂Sb₁.₈Te₃ stacks were cut into a rectangular slab with a size of ~15 × 2 × 2 mm³ and $\theta$ ~ 30°. Through microscopic analysis, we estimated the $t_{TE}$ value of our Co₂MnGa/Bi₀.₂Sb₁.₈Te₃ ATML to be 1.0 mm, resulting in the $R$ value of 0.43. This confirms the fabrication of Co₂MnGa/Bi₀.₂Sb₁.₈Te₃ ATML with the $\theta$ and $R$ values close to the optimal conditions for $z_{OD}T$, as depicted in Fig. 2d.

To check the thermoelectric transport property in our polycrystalline Co₂MnGa, we measured ANE using the plain Co₂MnGa slab (Supplementary Fig. 2). The anomalous Nernst coefficient of our Co₂MnGa is estimated to be 6.9 µV/K, which is comparable to the values in bulk single-crystalline Co₂MnGa[19,38] as well as polycrystalline Co₂MnGa[40,42]. The $\sigma$ ($\kappa$) value of our polycrystalline Co₂MnGa is almost same as (smaller than) that of single-crystalline Co₂MnGa[38].

### Lock-in thermography measurement

To investigate the transverse thermoelectric conversion processes in our ATML, LIT measurements were conducted (see Methods). The LIT technique based on the infrared thermometry enables the visualization of the temporal response and spatial distribution of temperature changes induced by an external periodic input with high temperature (< 0.1 mK) and spatial (around 10–20 µm) resolutions[43]. In LIT measurements, heating or cooling signals harmonically oscillating at the frequency of the periodic input signal are extracted as thermal images[37,44–48]. In this study, a square-wave-modulated charge current with the frequency $f$ (= 0.2–10.0 Hz), amplitude $J_c$ (= 1 A), and zero offset was applied to the sample along its longitudinal direction ($x$-axis in Fig. 3a, i), which induces temperature changes due to the thermoelectric conversion as well as Joule heating. By extracting the first harmonic component of the thermal images, the contribution of the thermoelectric effects ($\propto J_c$) can be retrieved free from that of Joule heating ($\propto J_c^2$)[46,47]. The captured thermal images were transformed into lock-in amplitude $A$ and phase $\phi$ images through Fourier analysis, where the $A$ ($\phi$) image shows the distribution of the magnitude (sign) of the induced temperature modulation and the $\phi$ image also gives

information on the time delay of the temperature modulation due to thermal diffusion. We carried out the LIT measurements in two configurations: cross-section (Fig. 3a) and top-side configurations (Fig. 3i).

### Visualization of off-diagonal Peltier effect in Co₂MnGa-based artificially tilted multilayer

Figure 3 shows the results of the LIT measurements for Co₂MnGa/Bi₀.₂Sb₁.₈Te₃ ATML in the absence of **H**. First, we examine the results for the cross-section configuration (Fig. 3a–h). The $A$ and $\phi$ images at $f$ = 10.0 Hz (top panels) and 0.2 Hz (bottom panels) are displayed in Fig. 3c, d, respectively. We can obtain information on the transient (nearly steady-state) temperature distribution induced by thermoelectric effects at $f$ = 10.0 Hz (0.2 Hz)[37,46]. At $f$ = 10.0 Hz, temperature modulation signals are obtained in the vicinity of the interfaces between the Co₂MnGa and Bi₀.₂Sb₁.₈Te₃ layers; the $A$ value reaches a maximum at the oblique junction interfaces and the 180° reversal of $\phi$ between the neighboring interfaces appears due to the Peltier effect. Importantly, in contrast to a Peltier-effect-induced temperature modulation at non-oblique junction interfaces[49,50], the $A$ signal is non-uniform along the oblique interfaces owing to the non-uniform $\mathbf{J_c}$ flow in ATML, which is the origin of the transverse thermoelectric conversion stemming from ODPE. At $f$ = 0.2 Hz, the cooling/heating signals are broadened through thermal diffusion, and the $A$ and $\phi$ signals exhibit zigzag-shaped patterns with an angle corresponding to a $\theta$ value of ~30°. As shown in the bottom panel of Fig. 3d, the $\phi$ value at the upper (lower) edge of the sample presents almost uniform values of -180° ( -0°) (see also the $x$-directional line profile shown in Fig. 3f), confirming transverse thermoelectric cooling and heating due to ODPE in Co₂MnGa/Bi₀.₂Sb₁.₈Te₃ ATML. Note that the $A$ value periodically changes along the $x$ direction because of the multilayer structure (Fig. 3e). The transverse thermoelectric cooling behavior can be clearly identified in the top-side configuration (Fig. 3i–n).

To evaluate the transverse thermoelectric cooling/heating performance of Co₂MnGa/Bi₀.₂Sb₁.₈Te₃ ATML, we averaged the LIT results over one Co₂MnGa/Bi₀.₂Sb₁.₈Te₃ unit. Figure 3g,h (3o,p) shows the $f$ dependence of the averaged $A$ and $\phi$ values, $A_{ave}$ and $\phi_{ave}$, for one Co₂MnGa/Bi₀.₂Sb₁.₈Te₃ unit in the cross-section (top-side) configuration, where the averaged areas are defined in Fig. 3c,d (3k,l). In both configurations, the $A_{ave}$ value monotonically increases and the $\phi_{ave}$

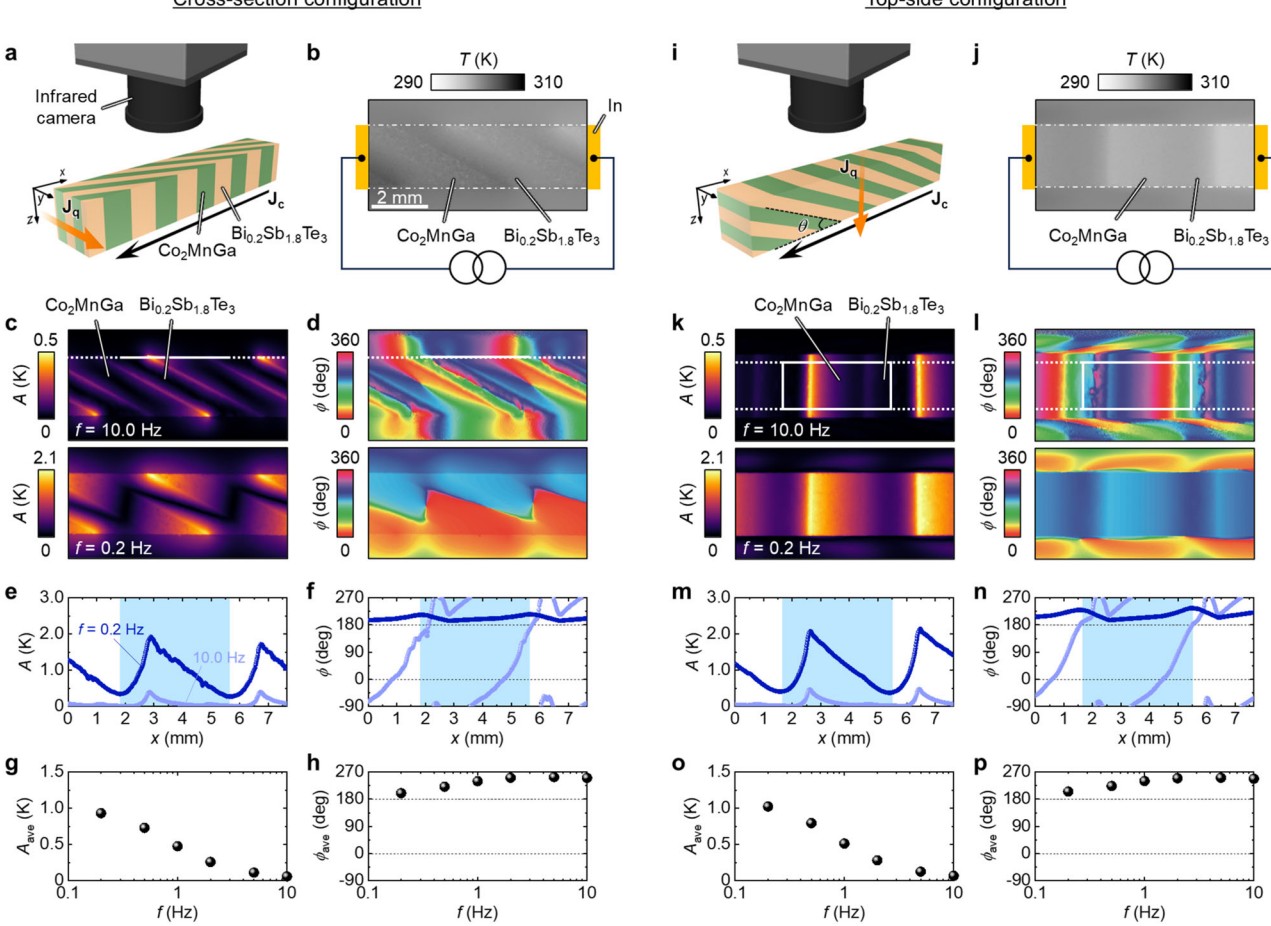

**Fig. 3 | Transverse thermoelectric conversion in $Co_2MnGa/Bi_{0.2}Sb_{1.8}Te_3$ ATML at zero magnetic field. a** Schematic of the sample structure in the cross-section configuration. **b** Steady-state temperature image during the LIT measurement in the cross-section configuration. **c**, **d** Lock-in amplitude $A$ (**c**) and phase $\phi$ (**d**) images at the lock-in frequency $f = 10.0$ and 0.2 Hz. **e**, **f** $x$-directional $A$ (**e**) and $\phi$ (**f**) profiles along the white dotted lines in the top panels of **c** and **d**, respectively. **g**, **h** $f$ dependence of the averaged lock-in amplitude $A_{ave}$ (**g**) and phase $\phi_{ave}$ (**h**) values over one $Co_2MnGa/Bi_{0.2}Sb_{1.8}Te_3$ unit. **i**–**p** Results for the top-side configuration. The $A_{ave}$ and $\phi_{ave}$ values in **g** and **h** (**o** and **p**) were estimated by averaging $A$ and $\phi$ signals in the areas defined by the white rectangles in **c** and **d** (**k** and **l**), corresponding to the blue shaded area in **e** and **f** (**m** and **n**), respectively. In all the LIT measurements, a square-wave-modulated charge current with an amplitude of 1 A and zero offset was applied.

value gradually gets closer to 180° with decreasing $f$, that is, approaching the temperature distribution in the steady state. The observed transverse thermoelectric temperature modulation is ~1 K at $f = 0.2$ Hz, which is much larger than that generated only by using OEE or AEE in a single conductor in a similar experimental condution[11,12,51].

**Transverse magneto-thermoelectric conversion in $Co_2MnGa$-based artificially tilted multilayer**

Next, we investigate the temperature modulation induced by transverse magneto-thermoelectric effects in $Co_2MnGa/Bi_{0.2}Sb_{1.8}Te_3$ ATML from the LIT images captured under an application of **H** with the magnitude of $H$. To hybridize ODPE and AEE, **H** was applied along the vertical direction ($z$ axis in Fig. 3a) [horizontal short direction ($y$ axis in Fig. 3i)] of the sample in the cross-section (top-side) configuration using electromagnets. The temperature change arising from the transverse magneto-thermoelectric effects, namely, AEE and/or OEE, exhibits an antisymmetric dependence with respect to the **H** direction, whereas the temperature change due to ODPE and/or the ordinary/ anisotropic magneto-Peltier effect showcases a symmetric dependence (note again that the contribution of the ordinary/anisotropic magneto-Peltier effect, the Onsager reciprocal of the ordinary/aniso-tropic magneto-Seebeck effect, is negligible in our samples). Thus, we

calculate $H$-odd-dependent LIT signals using the following equations:

$$A_{odd} = \left| A(+H)e^{-i\phi(+H)} - A(-H)e^{-i\phi(-H)} \right| / 2 \qquad (1)$$

$$\phi_{odd} = -\arg\left[ A(+H)e^{-i\phi(+H)} - A(-H)e^{-i\phi(-H)} \right] \qquad (2)$$

where $A(+H)$ [$\phi(+H)$] and $A(-H)$ [$\phi(-H)$] are defined as $A$ ($\phi$) measured in the positive ($-z$ or $+y$ direction) and negative ($+z$ or $-y$ direction) $H$, respectively. Figure 4a, b shows the $A_{odd}$ and $\phi_{odd}$ images for $Co_2MnGa/Bi_{0.2}Sb_{1.8}Te_3$ ATML at $\mu_0|H| = 0.8$ T and $f = 10.0$ Hz (top panels) and 0.2 Hz (bottom panels) in the cross-section configuration, where $\mu_0$ represents the vacuum permeability. The distribution of the $A_{odd}$ and $\phi_{odd}$ signals inside ATML is obviously different from that arising from ODPE at zero magnetic field (Fig. 3c, d). As highlighted in the images at $f = 10.0$ Hz, the $A_{odd}$ and $\phi_{odd}$ signals appear in the $Co_2MnGa$ regions. Importantly, despite their different distributions, the $H$-odd-dependent components contribute to the transverse thermoelectric cooling/heating with the same symmetry as ODPE (compare the thermal images at $f = 0.2$ Hz in Figs. 3d and 4b). Hereafter, we focus on the results for the top-side configuration because the signals at the sample edges in the cross-section

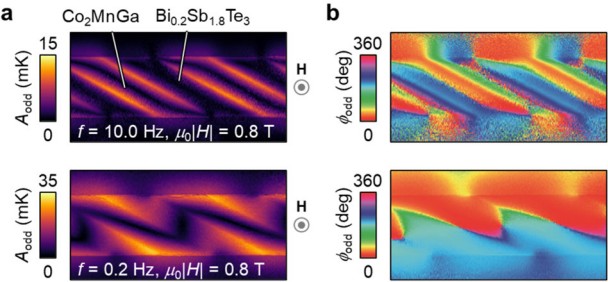

**Fig. 4 | Contribution of transverse magneto-thermoelectric effects in Co₂MnGa/Bi₀.₂Sb₁.₈Te₃ ATML in cross-section configuration. a, b** $H$-odd-dependent component of the lock-in amplitude $A_{odd}$ (**a**) and phase $\phi_{odd}$ (**b**) images at $\mu_0|H| = 0.8\,T$ and $f = 10.0$ and $0.2\,Hz$ in Co₂MnGa/Bi₀.₂Sb₁.₈Te₃ ATML. Here, the magnetic field **H** with the magnitude of $H$ was applied along the vertical direction ($z$ axis in Fig. 3a) of the sample and $\mu_0$ is the vacuum permeability.

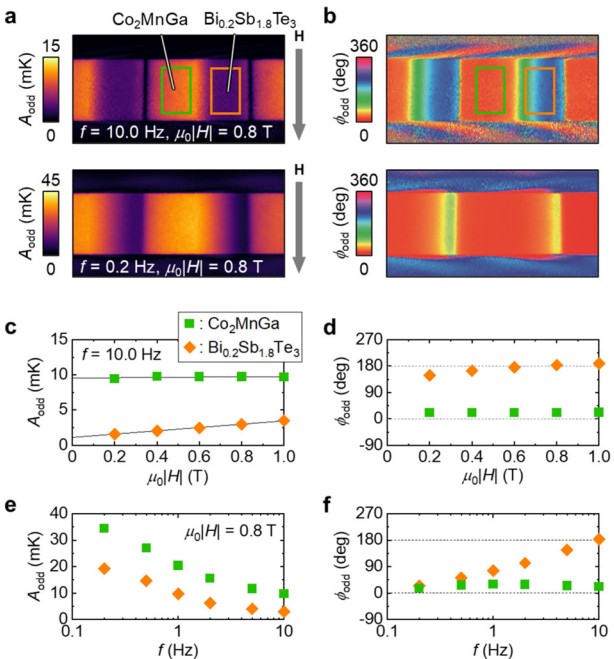

**Fig. 5 | Contribution of transverse magneto-thermoelectric effects in Co₂MnGa/Bi₀.₂Sb₁.₈Te₃ ATML in top-side configuration. a, b** $A_{odd}$ (**a**) and $\phi_{odd}$ (**b**) images at $\mu_0|H| = 0.8\,T$ and $f = 10.0$ and $0.2\,Hz$ in Co₂MnGa/Bi₀.₂Sb₁.₈Te₃ ATML. Here, **H** was applied along the horizontal short direction of the sample ($y$ axis in Fig. 3i). **c, d** $|H|$ dependence of $A_{odd}$ (**c**) and $\phi_{odd}$ (**d**) at $f = 10.0\,Hz$. Solid lines in **c** represent the results of linear fitting. **e, f** $f$ dependence of $A_{odd}$ (**e**) and $\phi_{odd}$ (**f**) at $\mu_0|H| = 0.8\,T$. The data points in **c–f** were obtained by averaging the temperature modulation signal in the area defined by the green and orange rectangles in **a** and **b** for the Co₂MnGa and Bi₀.₂Sb₁.₈Te₃ areas, respectively.

configuration are difficult to use for quantitative discussions due to the limitation of spatial resolution of LIT[37]. Figure 5a, b shows the $A_{odd}$ and $\phi_{odd}$ images for Co₂MnGa/Bi₀.₂Sb₁.₈Te₃ ATML at $\mu_0|H| = 0.8\,T$ and $f = 10.0\,Hz$ and $0.2\,Hz$ in the top-side configuration. At $f = 10.0\,Hz$, the $A_{odd}$ signals in the Co₂MnGa regions are greater than those in the Bi₀.₂Sb₁.₈Te₃ regions and the $\phi_{odd}$ signals are reversed between the Co₂MnGa (~0°) and Bi₀.₂Sb₁.₈Te₃ (~180°) regions. As shown in Fig. 5c, d, in the $\mu_0|H|$ range of 0.2–1.0 T, the $A_{odd}$ value is almost constant in the Co₂MnGa region with $\phi_{odd}$ ~ 0°, while the $A_{odd}$ value linearly increases with increasing $H$ in the Bi₀.₂Sb₁.₈Te₃ region with $\phi_{odd}$ ~ 180°. These tendencies are consistent with the $H$ dependence of the ANE signal in Co₂MnGa and that of the ONE signal in Bi₀.₂Sb₁.₈Te₃ (Supplementary

Fig. 2), indicating that the $A$ and $\phi$ images measured at non-zero $H$ contain the contributions of AEE in Co₂MnGa as well as OEE in Bi₀.₂Sb₁.₈Te₃. At lower $f$, the LIT images exhibit different temperature distributions as a result of superimposed contributions of AEE and OEE through thermal diffusion; the almost uniform $\phi_{odd}$ signal (~0°) across the entire sample area is observed at $f = 0.2\,Hz$ with the enhanced $A_{odd}$ signal. Figure 5e, f shows the $f$ dependence of the $A_{odd}$ and $\phi_{odd}$ values in the Co₂MnGa and Bi₀.₂Sb₁.₈Te₃ regions, respectively. With approaching the steady state, the $A_{odd}$ value monotonically increases in both the Co₂MnGa and Bi₀.₂Sb₁.₈Te₃ regions (Fig. 5e), while the $\phi_{odd}$ value remains ~0° in the Co₂MnGa region and gradually changes from ~180° to ~0° in the Bi₀.₂Sb₁.₈Te₃ region (Fig. 5f). These results indicate that the AEE-induced $J_q$ from the Co₂MnGa layer predominantly affects the temperature change even on the Bi₀.₂Sb₁.₈Te₃ surface, suggesting that ANE/AEE in Co₂MnGa modulates the transverse thermoelectric conversion in Co₂MnGa/Bi₀.₂Sb₁.₈Te₃ ATML in the steady state.

To verify our interpretation that the $A_{odd}$ and $\phi_{odd}$ signals stem from ANE in Co₂MnGa, we must distinguish the ANE contribution from the Seebeck-effect-driven anomalous Hall effect (AHE) observed in magnetic/thermoelectric hybrid systems[40,52,53], which can superimpose additional transverse thermopower onto ANE via the combination of the Seebeck effect in thermoelectric layers and AHE in magnetic layers. To this end, we performed the same LIT measurements for Co₂MnGa/Bi₂Te₃ ATML, in which $p$-type Bi₀.₂Sb₁.₈Te₃ was replaced by $n$-type Bi₂Te₃. Here, the sign of the contribution of Seebeck-effect-driven AHE should be reversed due to the different sign of $S_S$ between Bi₀.₂Sb₁.₈Te₃ and Bi₂Te₃. The fabrication procedure of ATML and the optimization conditions of ODSE for Co₂MnGa/Bi₂Te₃ ATML are the same as those for Co₂MnGa/Bi₀.₂Sb₁.₈Te₃ ATML (Methods and Supplementary Fig. 3). The results of the LIT measurements for Co₂MnGa/Bi₂Te₃ ATML are summarized in Supplementary Figs. 4–6. The direction of ODPE-induced $J_q$ is opposite and the magnitude of temperature modulation decreases, compared with that of Co₂MnGa/Bi₀.₂Sb₁.₈Te₃ ATML. This is because the sign (magnitude) of the difference in the Peltier coefficient ($= S_S T$) between the magnetic and thermoelectric components of ATMLs, $\Delta S_S T$ with $\Delta S_S$ being the difference in $S_S$, was reversed (reduced) by changing the sign of $S_S$ of the thermoelectric component from positive to negative. On the other hand, the $H$-odd component of the LIT signals in Co₂MnGa/Bi₂Te₃ ATML is comparable to that in Co₂MnGa/Bi₀.₂Sb₁.₈Te₃ ATML because of almost identical thermopower due to ONE between Bi₀.₂Sb₁.₈Te₃ and Bi₂Te₃ (Supplementary Table 1 and Supplementary Fig. 2). Since the sign of the transverse thermopower due to ANE in Co₂MnGa and Seebeck-effect-driven AHE in bilayers consisting of Co₂MnGa and $n$-type ($p$-type) thermoelectric materials is the same (opposite)[53,54], the transverse magneto-thermoelectric conversion in Co₂MnGa/Bi₂Te₃ ATML should be improved over that in Co₂MnGa/Bi₀.₂Sb₁.₈Te₃ ATML if Seebeck-effect-driven AHE contributes effectively to ATMLs. However, the observed $A_{odd}$ value in the Co₂MnGa region of Co₂MnGa/Bi₂Te₃ ATML is slightly smaller than that of Co₂MnGa/Bi₀.₂Sb₁.₈Te₃ ATML, which could be explained by electrical shunting in Bi₀.₂Sb₁.₈Te₃ and Bi₂Te₃. These results indicate that the contribution of Seebeck-effect-driven AHE on the total transverse magneto-thermopower is much smaller than that of ANE in our samples due to the unoptimized $\theta$ value and size ratio of the magnetic and thermoelectric layers for the magneto-thermoelectric effects (note that $\theta$ in our samples is designed for ODSE).

**Hybrid transverse magneto-thermoelectric conversion**
Here, we discuss the ANE-induced modulation of the total transverse thermoelectric conversion in our Co₂MnGa-based ATMLs. From an application viewpoint, the performance of the temperature modulation in the steady state is important; therefore, we focused on the results at the minimum $f$ (= 0.2 Hz). Figure 6a, b displays the $H$ dependence of the $A_{ave}$ and $\phi_{ave}$ signals at $f = 0.2\,Hz$ for Co₂MnGa/Bi₀.₂Sb₁.₈Te₃ and Co₂MnGa/Bi₂Te₃ ATMLs in the top-side

configuration. In both ATMLs, the magnitude of $A_{ave}$ exhibits the clear asymmetric $H$ dependence with a saturation behavior, which is consistent with the magnetization curve of Co$_2$MnGa in ATMLs (displayed in Fig. 6a, b), owing to the AEE contribution. Because of the dominant ODPE contribution and the positive (negative) $\Delta S_S$ for Co$_2$MnGa/

Bi$_{0.2}$Sb$_{1.8}$Te$_3$ (Co$_2$MnGa/Bi$_2$Te$_3$) ATML, an $H$-independent $\phi_{ave}$ value of -180° (-0°) appears in Co$_2$MnGa/Bi$_{0.2}$Sb$_{1.8}$Te$_3$ (Co$_2$MnGa/Bi$_2$Te$_3$) ATML. The $A_{ave}$ value is enhanced by applying the negative (positive) $H$ in Co$_2$MnGa/Bi$_{0.2}$Sb$_{1.8}$Te$_3$ (Co$_2$MnGa/Bi$_2$Te$_3$) ATML because the direction of generated $\mathbf{J}_q$ due to ODPE at zero $H$ is opposite to (the same as) that due to AEE at the positive $H$. The $H$-induced modulation ratio of the temperature change for Co$_2$MnGa/Bi$_{0.2}$Sb$_{1.8}$Te$_3$ and Co$_2$MnGa/Bi$_2$Te$_3$ ATMLs is estimated to be $|A_{ave}(+H) - A_{ave}(-H)|/A_{ave}(0\,T) \sim 5\%$ and 15% at $\mu_0|H| > 0.2$ T, respectively. The larger $H$-induced modulation ratio for Co$_2$MnGa/Bi$_2$Te$_3$ ATML is due to the smaller ODPE contribution [= $A_{ave}(0\,T)$]. These results reveal that AEE can exert a clear change in the average temperature modulation in ATMLs, although AEE appears only in the Co$_2$MnGa layers.

To quantitatively evaluate the thermoelectric performance based on the hybrid action of ODSE and ANE, we measured the transverse thermopower by applying a temperature difference $\Delta T$ between the top and bottom surfaces of Co$_2$MnGa-based AMTLs (see also Methods). Figure 7a, b shows the $H$ dependence of the transverse thermoelectric voltage $V_T$ at various values of $\Delta T$ for Co$_2$MnGa/Bi$_{0.2}$Sb$_{1.8}$Te$_3$ and Co$_2$MnGa/Bi$_2$Te$_3$ ATMLs, respectively. With increasing $\Delta T$, both the offset thermoelectric voltage due to ODSE and the $H$-induced change in the thermoelectric voltage due to ANE increase. Here, only the sign of the ODSE contribution is reversed between Co$_2$MnGa/Bi$_{0.2}$Sb$_{1.8}$Te$_3$ and Co$_2$MnGa/Bi$_2$Te$_3$ ATMLs because of the sign reversal of $\Delta S_S$. Figure 7c shows the temperature gradient $\nabla T$ dependence of the transverse thermoelectric field $E_T$ at $\mu_0 H = \pm 0.8$ T. The application of $H$ clearly changes the slope of the $E_T$-$\nabla T$ plots, i.e., the transverse thermopower $S_T$; the negative (positive) $H$ application enhances the magnitude of $S_T$ of Co$_2$MnGa/Bi$_{0.2}$Sb$_{1.8}$Te$_3$ (Co$_2$MnGa/Bi$_2$Te$_3$) ATML.

By combining the measured $S_T$ values and simulated $\sigma_{xx}$ and $\kappa_{yy}$ values (Fig. 2b, c and caption in Fig. 7), we estimate the figure of merit for hybrid transverse magneto-thermoelectric conversion $z_T T$ (= $S_T^2 \sigma_{xx} T / \kappa_{yy}$) at $T = 300$ K to be 0.088 and 0.095 (0.010 and 0.008) at +0.8 and −0.8 T, respectively, for Co$_2$MnGa/Bi$_{0.2}$Sb$_{1.8}$Te$_3$ (Co$_2$MnGa/Bi$_2$Te$_3$) ATML. These $z_T T$ values are several orders of magnitude larger than $zT$ for ANE, $z_N T$, in single magnetic materials ($< 7 \times 10^{-4}$ at 300 K). Most importantly, in our Co$_2$MnGa-based ATMLs, especially Co$_2$MnGa/Bi$_{0.2}$Sb$_{1.8}$Te$_3$ ATML, the $\mathbf{M}$-dependent change in $z_T T$ is also several times larger than $z_N T$. The reason why we achieve such a significant transverse magneto-thermoelectric modulation, surpassing the record-high $z_N T$, is owing to the presence of $\mathbf{M}$-independent ODSE. Since the $zT$ value is proportional to the square of the thermopower,

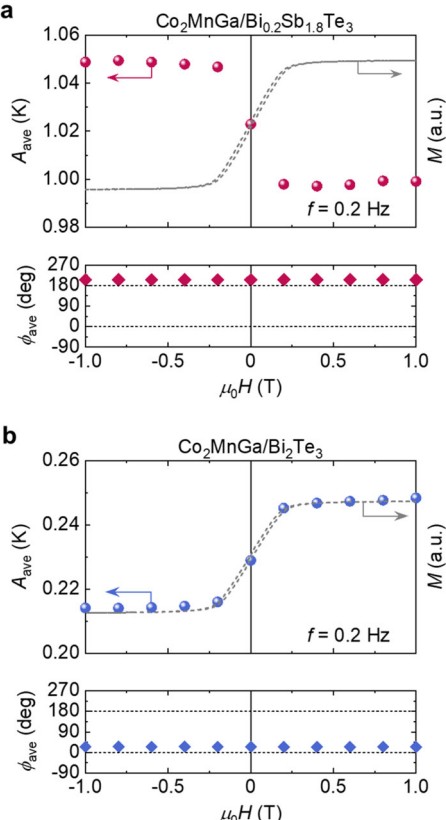

**Fig. 6 | Hybrid transverse magneto-thermoelectric temperature modulation in Co$_2$MnGa-based ATML.** $H$ dependence of the averaged $A_{ave}$ and $\phi_{ave}$ signals at $f = 0.2$ Hz for Co$_2$MnGa/Bi$_{0.2}$Sb$_{1.8}$Te$_3$ (**a**) and Co$_2$MnGa/Bi$_2$Te$_3$ (**b**) ATMLs. The magnetization $M$ curves of Co$_2$MnGa in each ATML measured using a vibrating sample magnetometer are also shown (note that Bi$_{0.2}$Sb$_{1.8}$Te$_3$ and Bi$_2$Te$_3$ are nonmagnetic).

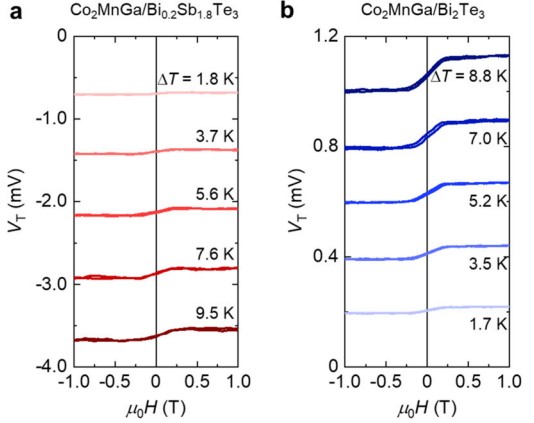
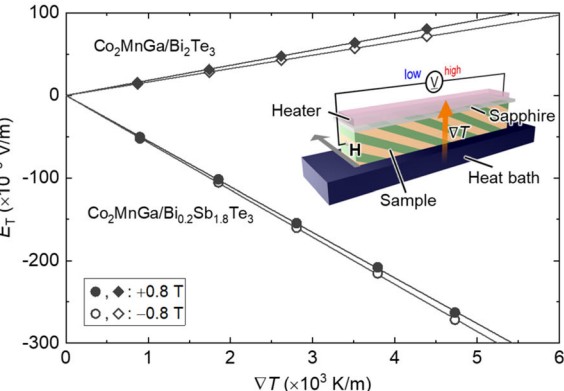

**Fig. 7 | Hybrid transverse magneto-thermoelectric generation in Co$_2$MnGa-based ATML. a, b** $H$ dependence of the transverse thermoelectric voltage $V_T$ at various $\Delta T$ values for Co$_2$MnGa/Bi$_{0.2}$Sb$_{1.8}$Te$_3$ (**a**) and Co$_2$MnGa/Bi$_2$Te$_3$ (**b**) ATMLs. $\Delta T$ denotes the temperature difference between the top and bottom surfaces of ATML. **c** $\nabla T$ dependence of the transverse electric field $E_T$ (= $V_T/l$) for Co$_2$MnGa/

Bi$_{0.2}$Sb$_{1.8}$Te$_3$ and Co$_2$MnGa/Bi$_2$Te$_3$ ATMLs at $\mu_0 H = \pm 0.8$ T. $\nabla T$ and $l$ denote temperature gradient and sample length, respectively. The transverse thermopower $S_T$, estimated by linear fitting, is −55.2 (+18.3) μV/K at +0.8 T and −57.1 (+16.2) μV/K at −0.8 T for Co$_2$MnGa/Bi$_{0.2}$Sb$_{1.8}$Te$_3$ (Co$_2$MnGa/Bi$_2$Te$_3$) ATML. The inset shows a schematic of the set-up for measuring the transverse thermopower.

$z_N T$ and $z_T T$ satisfy the following relations:

$$z_N T \propto S_N^2 \qquad (3)$$

$$z_T T \propto S_T^2 = \left(S_{OD} + \widetilde{S}_N\right)^2 = S_{OD}^2 + 2 S_{OD}\widetilde{S}_N + \widetilde{S}_N^2 \qquad (4)$$

where $\widetilde{S}_N$ is the effective anomalous Nernst coefficient considering the reduction due to the superimposition of ONE and the shunting effect in ATMLs. In ATMLs, the ANE contribution for $z_T T$ produces the additional $2 S_{OD}\widetilde{S}_N$ term of which magnitude depends on both $S_N$ and $S_{OD}$ (i.e., $S_T$ at $\mu_0 H = 0$ T), resulting that the magneto-thermoelectric modulation of $z_T T$ gets much greater than $z_N T$ owing to larger $|S_{OD}|$ than $|S_N|$. In fact, such a hybridization of ODSE and ANE is more prominent in Co$_2$MnGa/Bi$_{0.2}$Sb$_{1.8}$Te$_3$ ATML ($S_{OD} = -56.1$ μV/K) than in Co$_2$MnGa/Bi$_2$Te$_3$ ATMLs ($S_{OD} = +17.2$ μV/K). Thus, this unconventional synergy offers another route to develop physics and materials science on ANE.

## Discussion

Finally, we discuss prospects for advancing the ANE-hybridized transverse magneto-thermoelectric conversion in ATMLs. Exploring a magnetic material showing large $S_N$, which is the current mainstream research in spin caloritronics and topological materials science, is properly significant. If one can develop a material that exhibits twice larger $S_N$ than Co$_2$MnGa, for example, by introducing it in place of Co$_2$MnGa in Co$_2$MnGa/Bi$_{0.2}$Sb$_{1.8}$Te$_3$ ATML, the ANE-induced modulation of $z_T T$ reaches 0.01 at room temperature. In addition to ANE, the large magnitude of the base transverse thermopower at zero field due to ODSE is also indispensable, according to the results shown in the last section; a superior ODSE performance, e.g., larger $\Delta S_S$ in ATMLs, can enhance the magneto-thermoelectric modulation even if the contribution of ANE alone is the same. Supplementary Fig. 7a shows the dependence of the absolute value of $\widetilde{S}_N$ on the magneto-thermoelectric modulation of $z_T T$, i.e., $\Delta(z_T T) = \widetilde{S}_N^2 + 2|S_{OD}\widetilde{S}_N|$, at various magnitudes of $S_{OD}$. The presence of ODSE not only significantly improves the $\Delta(z_T T)$ value without changing $|\widetilde{S}_N|$ but also further increases the contribution of ANE on $\Delta(z_T T)$, i.e., the slope in Supplementary Fig. 7a, resulting in $\Delta(z_T T) \sim 0.05$ (0.13) at $|\widetilde{S}_N| = 7$ μV/K (20 μV/K) and $|S_{OD}| = 100$ μV/K at room temperature (note that over 100 μV/K of $|S_{OD}|$ has been reported in many ATMLs in previous studies[32–35,37]). Furthermore, as shown in Supplementary Fig. 7b, improving both $\widetilde{S}_N$ and $S_{OD}$ can enhance the $z_T T$ value, e.g., $z_T T \sim 0.35$ at $|\widetilde{S}_N| = 7$ μV/K and $|S_{OD}| = 100$ μV/K at room temperature. These findings emphasize the importance of investigating both ANE and ODSE (i.e., $S_N$ and $S_S$), which have been studied independently. Especially, the exploration of magnetic materials with large $S_S$ has received little attention so far, leading to new research on magnetic materials. Interface engineering in ATMLs is also promising since multilayering could increase $S_N$ itself intentionally via an interfacial spin-orbit interaction[55]. Further superimposition of the contribution of magneto-thermoelectric effects other than ANE is another route for improving the performance of transverse thermoelectric conversion in ATMLs. Since this study focuses on the demonstration of hybridizing ANE in ATMLs using a magnetic topological material, the contribution of ONE was not optimized; the opposite sign of $S_N$ in Bi$_{0.2}$Sb$_{1.8}$Te$_3$ or Bi$_2$Te$_3$ against $S_N$ in Co$_2$MnGa decreases the magneto-thermoelectric modulation. Optimizing the $\theta$ value and size ratio of magnetic and thermoelectric layers is also effective for further manifesting the contribution of Seebeck-effect-driven AHE in ATMLs. By appropriately designing these factors and including ODSE, ANE, ONE, Seebeck-effect-driven AHE, and other magneto-thermoelectric effects, dramatic improvements in the transverse magneto-thermoelectric conversion performance are possible in ATMLs.

In summary, we have demonstrated the giant ANE-induced modulation of transverse thermoelectric conversion that far exceeds the performance of ANE alone by fabricating ATMLs using a Weyl ferromagnet Co$_2$MnGa and thermoelectric materials. The infrared imaging technique based on LIT visualized the current-induced transverse thermoelectric cooling/heating behaviors. The LIT measurements under various $H$ values clarified the contributions of structure-induced ODPE, **H**-induced OEE, and **M**-induced AEE in Co$_2$MnGa-based ATMLs, proving the modulation of the transverse thermoelectric conversion performance due to hybridization of ODSE and ANE in Co$_2$MnGa. Owing to the existence of ODSE, the magneto-thermoelectric modulation of $z_T T$ in Co$_2$MnGa-based ATMLs induced by ANE was much greater than $z_N T$ in a single Co$_2$MnGa alloy that exhibits the record-high ANE performance at room temperature. The hybrid transverse magneto-thermoelectric conversion in ATMLs is potentially available even in the absence of magnetic fields; by utilizing permanent magnets with finite remanent magnetization as a magnetic component of ATML[37,56], not only ANE in the ferromagnetic layers but also ONE in the thermoelectric layers can be superimposed without external magnetic fields[57], eliminating the unnecessary electricity for magnetic field applications. The interdisciplinary fusion of transverse thermo-electrics by ANE and ODSE in ATMLs will stimulate research on thermoelectric applications in the topological materials science community more actively and open new directions for thermoelectric material exploration.

## Methods

### Preparation of Co$_2$MnGa, Bi$_{0.2}$Sb$_{1.8}$Te$_3$, and Bi$_2$Te$_3$ slabs and Co$_2$MnGa-based ATML

To prepare Co$_2$MnGa ingots, 99.97% purity Co, 99.99% purity Mn, and 99.9999% purity Ga shots (RARE METALLIC Co., Ltd.) were arc-melted with an atomic ratio of 2:1:1 in an Ar atmosphere. The resulting arc-melted ingot was homogenized in high vacuum at 1000 °C for 48 hours, followed by 600 °C for 72 hours. Subsequently, the ingot was crushed using planetary ball mill and sieved through a 100-μm mesh. The Co$_2$MnGa powder was subsequently sintered into a cylindrical slab with a diameter of 20 mm at 850 °C and a pressure of 30 MPa for 60 minutes in a vacuum chamber by the SPS method. Bi$_{0.2}$Sb$_{1.8}$Te$_3$ (Bi$_2$Te$_3$) ingots with a diameter of 20 mm and a height of 15 mm were prepared from 99.9% purity Bi-Sb-Te (Bi$_2$Te$_3$) powders, available from Toshima Manufacturing Co., Ltd, via the SPS method at 450 °C and 30 MPa for 60 minutes under the vacuum condition. The composition ratio of sintered Bi-Sb-Te slabs was confirmed to be Bi$_{0.2}$Sb$_{1.8}$Te$_3$ using scanning electron microscopy (SEM) with energy-dispersive X-ray spectroscopy (EDS) (Cross-Beam 1540ESB, Carl Zeiss AG). The polycrystalline nature of Co$_2$MnGa, Bi$_{0.2}$Sb$_{1.8}$Te$_3$, and Bi$_2$Te$_3$ alloys prepared by the above procedures were also obtained by SEM-EDS. For characterization of transport properties, the Co$_2$MnGa (Bi$_{0.2}$Sb$_{1.8}$Te$_3$ and Bi$_2$Te$_3$) ingot was cut into a rectangular slab with a size of ~10 × 1 × 1 mm$^3$ (~12 × 3 × 1 mm$^3$) using a diamond wire saw. Here, for the Bi$_{0.2}$Sb$_{1.8}$Te$_3$ and Bi$_2$Te$_3$ slabs, two rectangular slabs with different cutout directions were prepared (see Supplementary Note 2 and Fig. 8).

To fabricate Co$_2$MnGa-based ATMLs, the prepared cylindrical Co$_2$MnGa slab was sliced into many disks using the diamond wire saw. The SPS sintering for the Co$_2$MnGa/Bi$_{0.2}$Sb$_{1.8}$Te$_3$ and Co$_2$MnGa/Bi$_2$Te$_3$ stacks was carried out at 450 °C and 30 MPa for 60 minutes in the vacuum chamber. The elemental distribution of the stacks was characterized by SEM-EDS (Supplementary Fig. 1). Finally, the prepared Co$_2$MnGa/Bi$_{0.2}$Sb$_{1.8}$Te$_3$ and Co$_2$MnGa/Bi$_2$Te$_3$ stacks were cut into a rectangular slab of ATML with the diamond wire saw.

### Measurements of transport properties of single slab

The $\sigma$ and $S_S$ values and their $H$ dependence of single Co$_2$MnGa, Bi$_{0.2}$Sb$_{1.8}$Te$_3$, and Bi$_2$Te$_3$ slabs were measured using the system similar

to the Seebeck Coefficient/Electric Resistance Measurement System (ZEM-3, ADVANCE RIKO, Inc.)[58,59]. Here, the sample was clamped between two Cu blocks whose temperatures were independently controlled using ceramic heaters and Pt100 temperature sensors. **H** was applied along the short axis of the samples using an electromagnet. R-type (PtRh-Pt) thermocouple probes were attached to the sample to simultaneously measure the electric voltage and temperature difference. Simultaneous measurements of the generated thermoelectric voltage and temperature difference between the two thermocouple probes in the direction parallel to the applied $\nabla T$ through the Cu blocks enables accurate estimation of $S_S$. The $S_S$ value was quantified by fitting the temperature difference dependence of the thermoelectric voltage to a linear function. The magnitude of $\sigma$ was evaluated via the standard four-terminal method by applying a charge current through the Cu blocks. The $\kappa$ value was evaluated by multiplying the thermal diffusivity measured using the laser flash method, the specific heat using the differential scanning calorimetry, and the density using the Archimedes method. $S_N$ was measured using a homemade temperature gradient generator combined with an electromagnet through a method similar to that described in ref. 52 (see also Supplementary Fig. 3). The sample was bridged onto two surface-anodized Al blocks, where one of the Al blocks was thermally mounted on a heat bath and the other had a chip heater to apply $\nabla T$. The surface of the sample was coated with black ink and the magnitude of $\nabla T$ was measured with the infrared camera. While sweeping **H** in the direction perpendicular to $\nabla T$, generated $V_T$ in the direction perpendicular to both $\nabla T$ and **H** was measured. The $S_N$ value was estimated by fitting the $\nabla T$ dependence of $E_T$, i.e., $V_T$ divided by the sample width, with a linear function (Supplementary Fig. 3). All measurements were performed at room temperature and atmospheric pressure.

### LIT measurements

The LIT measurements were conducted with Enhanced Lock-In Thermal Emission (ELITE, DCG Systems G.K.) at room temperature and atmospheric pressure. For the LIT measurements under the vertical **H**, the ATML samples were directly mounted on the center of an electromagnet to ensure the uniform $H$. For the LIT measurements under the horizontal **H**, the ATML samples were securely mounted on a phenolic resin plate with low thermal conductivity to minimize the heat leakage through thermal conduction. To improve infrared emissivity and ensure uniform emission properties during the LIT measurements, the top and side surfaces of the samples were coated with insulating black ink whose emissivity is > 0.94 (JSC-3, JAPANSENSOR Corporation). The minimum $f$ value during the LIT measurements was determined by considering the thermal boundary conditions; we chose 0.2 Hz as a minimum $f$ because we have confirmed that there is no influence of parasitic signals, such as the Peltier effect at the sample edges due to wiring, to the signal on the sample in the viewing area of LIT at 0.2 Hz. The viewing area of all the lock-in images is $512 \times 260$ pixels = $7.68 \times 3.90$ mm$^2$.

### Measurements of transverse thermopower in Co$_2$MnGa-based ATMLs

A schematic of the measurement set-up for the transverse thermopower in Co$_2$MnGa-based ATMLs is shown in the inset of Fig. 7c. The ATML samples for thermopower measurements were fixed on a heat bath made of a surface-anodized Al block. To apply $\nabla T$ to the samples, a chip heater was attached to the top surface of the sample and a sapphire substrate was inserted between the heater and sample to ensure uniform temperature gradient application. The $\nabla T$ value was determined using the infrared camera by coating the side surface of the sample with the black ink. During sweeping **H** in the width direction of the samples perpendicular to $\nabla T$, $V_T$ in the longitudinal direction was measured. Finally, the $S_T$ value was quantified by linear fitting of $E_T$-

$\nabla T$ plots, where $E_T = V_T/l\nabla T$ with $l$ representing the longitudinal length of the sample.

## Data availability

The data generated in this study are provided in the Supplementary Information/Source Data file. Source data are provided with this paper.

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

## Acknowledgements

The authors thank T. Seki, Y. Sakuraba, W. Zhou, F. Makino, K. Suzuki, and M. Isomura for technical support and valuable discussions. This work was partially supported by ERATO "Magnetic Thermal Management Materials Project" (No. JPMJER2201) from JST, Japan; Grant-in-Aid for Scientific Research (S) (No. 22H04965) from JSPS KAKENHI, Japan; and NEC Corporation (K.U.).

## Author contributions

T.H. and K.U. planned and designed the experiments, prepared the samples, and performed the LIT experiments. T.H. analyzed the LIT data, measured the thermoelectric, electric, thermal transport properties of the samples, and developed the explanation of the experimental results. K.U. collected magnetic properties of ATMLs. F.A. analytically simulated the transport properties in ATMLs. H.S.A conducted the microstructural characterization. T.H. and K.U. wrote the manuscript. K.U. supervised the project. All authors discussed the results and commented on the manuscript.

## Competing interests

The authors declare no competing interests.
