## [Peer Review file · Nature Communications]

Hybridizing anomalous Nernst effect in artificially tilted multilayer based on magnetic topological material

Corresponding Author: Professor Ken-ichi Uchida

Version 0:

Reviewer comments:

Reviewer #1

(Remarks to the Author)

The authors realized a large modulation of anomalous Nernst effect (ANE) by integrating magnetic topological materials into artificially tilted multilayers (ATMLs), which provides new insights for spin caloritronics. However, it seems that the explanation on why this enhancement can be achieved is not very clear. It would be better the authors can clarify the following issues.

1. By adopting materials with large difference in S_s , σ and κ to form periodic structures, the idea of ATMLs looks very similar to the concept of 1D phononic crystal. Is there any guidelines as how the structure is designed? For example, the choose of materials, as well as their thicknesses, the tilted angle, etc. Could we adopt a similar bandstructure-like theory to quantitatively describe ATMLs?
2. Following the last comment, is the number of repeating period matters? Or in other words, the boundaries or surfaces play a role in the measurements?
3. The authors show very nice schematics for ATMLs. Are there any experimental evidences (such as cross-sectional electron microscope images) to support the sharp repeating interfaces of CMG/BST?
4. It is still not very clear how is the modulation of ANE achieved. Is it an interface effect that arises from the phonon scattering at the CMG/BST interface?
5. Does the polycrystalline nature of the sample plays a role, e.g. the lateral phonon scattering from the grain boundary?
6. A minor issue is that how the low ac frequency is chosen? If we use a very high frequency, would the change of phonon penetration depth have an influence on the results?

Reviewer #2

(Remarks to the Author)

In this manuscript, the authors introduce artificially tilted multilayers (ATMLs) composed of magnetic topological and thermoelectric materials to realize transverse thermoelectric cooling. The non-uniform charge carriers passed through the tilted structures can be converted into orthogonal heat currents, which provides another pathway to study the transverse thermoelectric effect. As the authors argued, although the hybrid transverse magneto-thermoelectric conversion based on ONE/OEE and ODSE/ODPE in ATMLs has recently been reported by the same group, hybridizing ANE/AEE has not been demonstrated and magnetic topological materials have not been incorporated into AMTLs so far. The proposed ANE/AEE in ATMLs will provide new ideas for broadening the research category of transverse thermoelectrics. Therefore, the manuscript can be considered for publication in Nature Communications after some necessary revisions.

1. In the previous work (ref. 36), the tilted multilayers that include $\text{Bi}_{88}\text{Sb}_{12}$ and $\text{Bi}_{0.2}\text{Sb}_{1.8}\text{Te}_3$ pair have been reported to exhibit a magnetically enhanced hybrid transverse thermoelectric cooling effect. Could the ANE/AEE also be realized in the $\text{Bi}_{88}\text{Sb}_{12}$ and $\text{Bi}_{0.2}\text{Sb}_{1.8}\text{Te}_3$ pair? If so, the difference and novelty of the currently proposed $\text{Co}_2\text{MnGa}/\text{Bi}_{0.2}\text{Sb}_{1.8}\text{Te}_3$ ATMLs should be further highlighted, compared with the previously reported $\text{Bi}_{88}\text{Sb}_{12}$ and $\text{Bi}_{0.2}\text{Sb}_{1.8}\text{Te}_3$ pair.
2. The authors clarified the synthesized " Co_2MnGa , $\text{Bi}_{0.2}\text{Sb}_{1.8}\text{Te}_3$, and Bi_2Te_3 are polycrystalline, suggesting their isotropic transport properties." However, Bi_2Te_3 is a famous Seebeck thermoelectric material and has been proven to exhibit anisotropic transport properties. Therefore, some necessary characterizations could be needed to support the argument.
3. As stated by the authors, the Bi-Sb-Te powders were piled up and bonded together with the Co_2MnGa discs. Thus, will

- the Bi-Sb-Te powders diffuse into the Co₂MnGa matrix material, thereby affecting its transverse thermoelectric properties?
4. The data of specific heat using the differential scanning calorimetry should be provided.
 5. In Fig. 3 and Fig. 4, the direction of infrared camera imaging and the configuration of testing should be clearly marked in the diagram. Moreover, it is preferable to label the material composition and temperature changes of each region in the thermal imaging image one-to-one in the image (Fig. 3c,d,k,i).
 6. What are the temperature and spatial resolutions of the LIT technique?

Some minor issues are worth considering for improving this manuscript's readability.

1. The manuscript focuses on the cooling phenomena of thermoelectric materials (whether longitudinal or transverse), should the title highlight the cooling-related effects more prominently?
2. On lines 28 and 29, "...a thermoelectric module based on the Seebeck/Peltier effect usually consists of a bunch of thermocouples, made of p-type and n-type...", where the "thermocouples" used here seems to be misplaced.
3. On line 36, "Separating the J_c and J_q directions not only reduces the number of electrodes but also enhances the...". For both Seebeck effect-based and Nernst effect-based devices, when the number of thermoelectric legs in the device is the same, the number of electrodes seems to be the same.
4. On lines 60 and 61, "a large transverse conversion by ODSE/ODPE has been realized using conventional thermoelectric materials.", where relevant references may be needed.
5. On line 112, The simulation details related to θ and R may need to be provided.
6. On line 116, "...well as polycrystalline Co₂MnGa[39,41].", Co₂MnGa in reference [39] is reported as a film while in reference [41] it is polycrystalline bulk, the latter could be more comparable to the currently studied Co₂MnGa as they are all fabricated using the SPS method.

Reviewer #3

(Remarks to the Author)

This study investigates the anomalous Nernst effect (ANE) and the off-diagonal Seebeck effect (ODSE) in artificially tilted multilayers (ATMLs). Transverse thermoelectric effects such as ANE have attract significant attention due to their potential to reduce device dimensions in thermoelectric generators. The authors employ a large ANE in topological materials and ODSE in ATMLs cooperatively to enhance transverse thermoelectric performance. This approach is novel. They present high-quality data on charge-heat current conversion and vice versa and discuss an intriguing hybrid action between ODSE and ANE in ATMLs. The manuscript is well-written, and the scientific discussion is robust. However, its impact appears limited for general readers due to the following reasons:

1. Modulation of the transverse thermoelectric effect by magnetic fields is small as seen in Fig. 6c.
2. The experimental design and results in Figs. 1-4 closely resemble those in their recent work (K. Uchida et al., Adv. Energy Mater. 2024, 14, 2302375), where they had reported high thermoelectric performance in ATMLs.

In conclusion, while the perspective, data, and discussion quality are excellent, the results lack significant impact and do not meet the standards expected by Nature Communications. This manuscript might find better reception in journals specializing in applied physics or material science, where it could benefit other researchers.

Minor comment: The sentence in Line 132, "During the LIT measurements, an in-plane H with the magnitude of H was applied along the short axis of the sample (y axis in Fig. 3a,i) using an electromagnet," should be relocated since no H field was applied in Fig. 3.

Version 1:

Reviewer comments:

Reviewer #1

(Remarks to the Author)

The authors have addressed all my previous concerns in a satisfactory way. I recommend the publication of the revised manuscript.

Reviewer #2

(Remarks to the Author)

The authors have made satisfactory revision. It can be acceptable as is.

Reviewer #3

(Remarks to the Author)

I appreciate the authors' effort. The revised manuscript clearly explains the importance of the hybridization effect, as shown in Supplementary Fig. 6. However, I still do not fully agree with the authors' response.

Regarding the first point (Modulation of the transverse thermoelectric effect by magnetic fields is small), I find the authors' rebuttal to be reasonable. I now understand that $\Delta z_{TT} \sim 0.05$ under ideal conditions is too significant to ignore, and this result cannot be achieved solely through ANE/AEE. This is likely to attract attention from the community. However, I am

concerned that this approach cannot reverse the output sign through magnetization reversal, thus losing the advantage of ANE/AEE. I would appreciate further comments from the authors on this.

As for the second point(The experimental design and results in Figs. 1-4 closely resemble those in their recent work), I feel the rebuttal is somewhat insufficient. To further distinguish this work from the authors' previous work, would it be possible to include a visualization of the magnetization-dependent transverse thermoelectric signal in the cross-sectional configuration?

Version 2:

Reviewer comments:

Reviewer #3

(Remarks to the Author)

I appreciate the efforts of the authors. Regarding the first point, I now have a good understanding. For the second point, I believe the additional data has made the discussion more comprehensive. While I have some reservations about the overall impact of this paper, it represents a valuable contribution to the field. Therefore, I recommend it for the editor's careful consideration.

[Reply to the comments of Reviewer #1]

We are grateful to Reviewer #1 for their comments, which were very helpful to us in improving our manuscript. Our responses to your individual comments are provided below.

(Comment 1)

The authors realized a large modulation of anomalous Nernst effect (ANE) by integrating magnetic topological materials into artificially tilted multilayers (ATMLs), which provides new insights for spin caloritronics. However, it seems that the explanation on why this enhancement can be achieved is not very clear. It would be better the authors can clarify the following issues.

(Answer 1)

Thank you very much for your helpful comments and constructive evaluation. In accordance with your comments, we have carefully revised the manuscript. The authors' reply to the comments and revised parts of the manuscript are detailed below.

(Comment 1-1)

By adopting materials with large difference in S_s , σ and κ to form periodic structures, the idea of ATMLs looks very similar to the concept of 1D phononic crystal. Is there any guidelines as how the structure is designed? For example, the choose of materials, as well as their thicknesses, the tilted angle, etc. Could we adopt a similar bandstructure-like theory to quantitatively describe ATMLs?

(Answer 1-1)

Thank you very much for pointing this out. Indeed, several transport and structural parameters are involved in optimizing thermoelectric transport performance of artificially tilted multilayers (ATMLs) due to the off-diagonal Seebeck effect (ODSE). However, in ATMLs with mm- and sub-mm-order dimensions used in this study, the transverse thermoelectric performance of ATMLs is dominantly determined by macroscale transport properties of constitute materials, which can be quantitatively estimated by analytical matrix calculations based on Goldsmid's method [ref. 30 in the revised manuscript: Goldsmid, H. J., *J. Electron. Mater.* **40**, 1254–1259 (2011)]. In this study, following this simulation, we determined the material combination, thickness of each material, and tilt angle of ATMLs. To make this clear for readers, we have added the details of analytical calculation in Supplementary Note 1 with citation on Line 101 in the main text.

(Authors' additional comment)

Although the thermoelectric performance of our system is determined by the macroscale parameters, it is known that anisotropic crystals and superlattices, whose c -axis is inclined at a non-zero angle relative to the substrate surface normal, also exhibit the transverse thermoelectric conversion due to ODSE [e.g., Lengfellner, H. et al., *Appl. Phys. Lett.* **60**, 501 (1992), Takahashi, K. et al., *Appl. Phys. Lett.* **97**, 021906 (2010), and Tang, Y. et al., *J. Electron. Mater.* **44**, 2095 (2015)]. In such systems, the anisotropic band structures importantly affect their transport properties in a similar manner to phononic crystals.

(Comment 1-2)

Following the last comment, is the number of repeating period matters? Or in other words, the boundaries or surfaces play a role in the measurements?

(Answer 1-2)

As discussed in Answer 1-1, the transverse thermoelectric performance of our ATMLs is dominantly determined by macroscale transport properties of constitute materials, and the effect of the repeating period number on the band structures can be excluded. Nevertheless, the measured transverse thermoelectric performance due to ODSE in ATMLs can be affected by boundaries and surfaces for the following reason. Namely, as the repetition number of layers, i.e., the interface density, decreases without changing the total sample size and thickness ratio of ATMLs, the anisotropic flow of heat and charge currents becomes blurred due to the boundary condition, making the measured transverse thermoelectric performance due to ODSE more distant to the theoretical value [e.g., Kanno T. et al., *Appl. Phys. Lett.* **101**, 011906 (2012)]. This is one reason why the measured dimensionless figure of merit (Line 262) is slightly deviated from the calculated value (Line 106). However, this small effect of the boundary condition does not affect the conclusion of our study. Note that the thickness of the layers in our ATMLs (~0.7 mm) is the nearly minimum thickness that can be cutout as a disk with sharp parallel surfaces in this study.

(Comment 1-3)

The authors show very nice schematics for ATMLs. Are there any experimental evidences (such as cross-sectional electron microscope images) to support the sharp repeating interfaces of CMG/BST?

(Answer 1-3)

Thank you very much for pointing this out. In accordance with your comment, we performed the cross-sectional scanning electron microscopy (SEM) with energy-dispersive X-ray spectroscopy (EDS). Supplementary Fig. 1a,b shows the low- and high-magnification SEM-EDS images and corresponding line profiles for the $\text{Co}_2\text{MnGa}/\text{Bi}_{0.2}\text{Sb}_{1.8}\text{Te}_3$ multilayer, respectively. Here, we used a rectangular sample comprising the $\text{Co}_2\text{MnGa}/\text{Bi}_{0.2}\text{Sb}_{1.8}\text{Te}_3$ stacks in which the cut angle was perpendicular to the stacking plane (i.e., the tilt angle $\theta = 90^\circ$). The results identify the sharp repeating boundaries with little atomic interdiffusion in $\text{Co}_2\text{MnGa}/\text{Bi}_{0.2}\text{Sb}_{1.8}\text{Te}_3$ ATMLs. The similar results were obtained in the $\text{Co}_2\text{MnGa}/\text{Bi}_2\text{Te}_3$ multilayers as shown in Supplementary Fig. 1c,d.

To explain the above experiments, in the revised manuscript, we have added sentences on Lines 111-114 and an author (Hossein Sepehri-Amin) in light of the contribution to these experiments.

(Comment 1-4)

It is still not very clear how is the modulation of ANE achieved. Is it an interface effect that arises from the phonon scattering at the CMG/BST interface?

(Answer 1-4)

Relating to Answers 1-1 and 1-2, the thermopower due to ANE itself, i.e. the anomalous Nernst coefficient, in the sub-mm-thick Co_2MnGa layers is not modulated by an interface effect in our ATMLs. However, the contribution of ANE to the total transverse thermoelectric performance in ATMLs is boosted by the hybrid action of ANE and ODSE, as shown in Equations (3) and (4) in the revised manuscript.

To make it clearer, we have added the explanation of this hybrid action on Lines 288-295 in the revised manuscript with adding Supplementary Fig. 6.

On the other hand, the previous study reveals that ANE itself could be enhanced by the interfacial spin-orbit interaction in spintronic multilayers [Uchida, K. et al. *Phys. Rev. B* **92**, 094414 (2015)], and interface engineering could allow to increase ANE itself intentionally even in ATMLs. Thus, in the revised manuscript, we described this point as a future outlook (Lines 297-299).

(Comment 1-5)

Does the polycrystalline nature of the sample play a role, e.g. the lateral phonon scattering from the grain boundary?

(Answer 1-5)

As you pointed out, the polycrystalline structure with a number of grain boundaries may increase the probability of phonon scattering in general. In our case, even though the electrical conductivity of single-crystalline Co₂MnGa and our polycrystalline Co₂MnGa is almost same (7.8×10^5 and 8.0×10^5 S/m, respectively), the thermal conductivity of our polycrystalline Co₂MnGa (19 W/mK) is smaller than that of single-crystalline Co₂MnGa (24 W/mK) [see ref. 38 in revised manuscript: Guin S. N. et. al. *NPG Asia Mater.* **11**, 16 (2019) for the values for single-crystalline Co₂MnGa]. This may suggest that the polycrystalline nature of our Co₂MnGa affect the lattice thermal conductivity. However, partly because it is difficult to compare transport properties between single crystal and polycrystal with the same composition for Bi-Sb-Te and Bi-Te, we cannot estimate how the polycrystalline nature improves the thermoelectric performance in our ATMLs quantitatively. Therefore, in the revised manuscript, we have only described the difference in electrical and thermal conductivity between the polycrystalline and single-crystalline Co₂MnGa samples on Lines 122-123.

(Comment 1-6)

A minor issue is that how the low ac frequency is chosen? If we use a very high frequency, would the change of phonon penetration depth have an influence on the results?

(Answer 1-6)

The frequency of lock-in thermography (LIT) measurements is generally determined by taking into account thermal boundary conditions to investigate the behavior of thermal diffusion. Although LIT measurements at lower frequencies can give us information on nearly steady-state temperature distribution, the measurement data are more susceptible to parasitic signals originating from thermal boundary conditions, such as the Peltier effect at sample edges due to wiring. The reason why we chose 0.2 Hz as a minimum frequency is that a typical minimum frequency for LIT measurements is around 0.1 Hz and we have confirmed that there is no influence of signals from the sample edge at 0.2 Hz. In contrast, if we perform LIT measurements at a frequency higher than the maximum frequency used in this study (= 10 Hz), we can visualize more localized temperature distribution (note that a maximum frequency of LIT is typically 50-100 Hz due to the limitation of a frame rate of an infrared camera). However, it is difficult to discuss a microscopic phonon contribution because of the slow time scale and limited spatial resolution of LIT (~20 μ m), which is much larger than typical values of a phonon penetration depth, at least a few hundred nanometers. LIT measurements at 10 Hz are sufficient to discuss the behavior of thermal diffusion in our ATMLs with mm-scale dimensions.

In response to Comment 1-6, we have added the sentences on Line 129 and Lines 380-384.

[Reply to the comments of Reviewer #2]

We are grateful to Reviewer #2 for their comments, which were very helpful to us in improving our manuscript. Our responses to your individual comments are provided below.

(Comment 2)

In this manuscript, the authors introduce artificially tilted multilayers (ATMLs) composed of magnetic topological and thermoelectric materials to realize transverse thermoelectric cooling. The non-uniform charge carriers passed through the tilted structures can be converted into orthogonal heat currents, which provides another pathway to study the transverse thermoelectric effect. As the authors argued, although the hybrid transverse magneto-thermoelectric conversion based on ONE/OEE and ODSE/ODPE in ATMLs has recently been reported by the same group, hybridizing ANE/AEE has not been demonstrated and magnetic topological materials have not been incorporated into AMTLs so far. The proposed ANE/AEE in ATMLs will provide new ideas for broadening the research category of transverse thermoelectrics. Therefore, the manuscript can be considered for publication in Nature Communications after some necessary revisions.

(Answer 2)

Thank you very much for your helpful comments and positive evaluation. In accordance with your comments, we have carefully revised the manuscript. The authors' reply to the comments and revised parts of the manuscript are detailed below.

(Comment 2-1)

In the previous work (ref. 36), the tilted multilayers that include $\text{Bi}_{88}\text{Sb}_{12}$ and $\text{Bi}_{0.2}\text{Sb}_{1.8}\text{Te}_3$ pair have been reported to exhibit a magnetically enhanced hybrid transverse thermoelectric cooling effect. Could the ANE/AEE also be realized in the $\text{Bi}_{88}\text{Sb}_{12}$ and $\text{Bi}_{0.2}\text{Sb}_{1.8}\text{Te}_3$ pair? If so, the difference and novelty of the currently proposed $\text{Co}_2\text{MnGa}/\text{Bi}_{0.2}\text{Sb}_{1.8}\text{Te}_3$ ATMLs should be further highlighted, compared with the previously reported $\text{Bi}_{88}\text{Sb}_{12}$ and $\text{Bi}_{0.2}\text{Sb}_{1.8}\text{Te}_3$ pair.

(Answer 2-1)

Thank you very much for pointing this out. In ref. 37 in the revised manuscript (ref. 36 in the original manuscript), the *ordinary* Nernst/Ettingshausen effect (ONE/OEE) in $\text{Bi}_{88}\text{Sb}_{12}$ was used for the hybrid transverse magneto-thermoelectric cooling and the *anomalous* Nernst/Ettingshausen effect (ANE/AEE) cannot be manifested in $\text{Bi}_{88}\text{Sb}_{12}/\text{Bi}_{0.2}\text{Sb}_{1.8}\text{Te}_3$ artificially tilted multilayers (ATMLs) because both $\text{Bi}_{88}\text{Sb}_{12}$ and $\text{Bi}_{0.2}\text{Sb}_{1.8}\text{Te}_3$ are nonmagnetic. We would like to emphasize that hybridizing ANE/AEE into ATMLs is demonstrated in this study for the first time.

In the revised manuscript, to make the difference in the previous reports clearer, we have added the sentence on Lines 49-50 and 65-69.

(Comment 2-2)

The authors clarified the synthesized " Co_2MnGa , $\text{Bi}_{0.2}\text{Sb}_{1.8}\text{Te}_3$, and Bi_2Te_3 are polycrystalline, suggesting their isotropic transport properties." However, Bi_2Te_3 is a famous Seebeck thermoelectric material and has been proven to exhibit anisotropic transport properties. Therefore, some necessary characterizations could be needed to support the argument.

(Answer 2-2)

We appreciate your constructive comment. To investigate the anisotropy of transport properties, we performed additional experiments as described below. First, we newly sintered large cylindrical Bi_2Te_3 and $\text{Bi}_{0.2}\text{Sb}_{1.8}\text{Te}_3$ ingots with a diameter of 20 mm and a height of 15 mm. Then, as shown in Supplementary Fig. 7a, two rectangular slabs, named “Slab 1” and “Slab 2”, with a size of $\sim 12 \times 3 \times 1 \text{ mm}^3$ were cut out from the same ingot using a diamond wire saw, where the longitudinal direction of Slab1 (Slab2) was along the diameter (height) direction of the ingot, i.e., perpendicular (parallel) to the pressing direction during spark plasma sintering (SPS). We should note that, in the original manuscript, we measured the transport properties of the sample cut out as in Slab 1. Supplementary Fig. 7b shows the Seebeck coefficient S_S and electrical conductivity σ of Slabs 1 and 2 for Bi_2Te_3 and $\text{Bi}_{0.2}\text{Sb}_{1.8}\text{Te}_3$. In both the Bi_2Te_3 and $\text{Bi}_{0.2}\text{Sb}_{1.8}\text{Te}_3$ slabs, there was little difference in S_S between Slabs 1 and 2, while the magnitude of σ of Slab 2 was smaller than that of Slab 1 even though our Bi_2Te_3 and $\text{Bi}_{0.2}\text{Sb}_{1.8}\text{Te}_3$ slabs are polycrystalline. The behavior of isotropic S_S and anisotropic σ in Bi-Te and Bi-Sb-Te prepared by SPS has been also reported in the previous study; it is not due to their intrinsic band structure but due to the particle shape of the matrix powder [Kim, D.-H et al, *Acta Mater.* **59**, 405 (2011)]. However, we would like to emphasize that the observed anisotropy of σ has little impact on the transport properties of our ATMLs; the value of dimensionless figure of merit for the off-diagonal Seebeck effect calculated with the parameters for Slab 1 was almost the same as that for Slab 2 (see Supplementary Fig. 7c,d and Supplementary Note 1 for calculation details).

In the revised manuscript, we have added above experimental results in Supplementary Information (Note 2 and Fig. 7) to introduce the characteristics of sintered Bi_2Te_3 and $\text{Bi}_{0.2}\text{Sb}_{1.8}\text{Te}_3$. For the analytical calculation (Fig. 2 and Supplementary Fig. 3), we used the values for Slab 1.

Again, thank you so much for your insightful comments making us aware of this anisotropic behavior of electrical conductivity in sintered samples.

(Comment 2-3)

As stated by the authors, the Bi-Sb-Te powders were piled up and bonded together with the Co_2MnGa discs. Thus, will the Bi-Sb-Te powders diffuse into the Co_2MnGa matrix material, thereby affecting its transverse thermoelectric properties?

(Answer 2-3)

Thank you very much for your insightful comments. In accordance with your comment, we performed the cross-sectional scanning electron microscopy (SEM) and energy-dispersive X-ray spectroscopy (EDS). Supplementary Fig. 1b shows the high-magnification SEM-EDS images and corresponding line profiles for the $\text{Co}_2\text{MnGa}/\text{Bi}_{0.2}\text{Sb}_{1.8}\text{Te}_3$ multilayers, respectively. Here, we used a rectangular sample comprising the $\text{Co}_2\text{MnGa}/\text{Bi}_{0.2}\text{Sb}_{1.8}\text{Te}_3$ stacks in which the cut angle was perpendicular to the stacking plane (i.e., the tilt angle $\theta = 90^\circ$). The results identify that the interdiffusion is limited to a region of $\sim 10 \mu\text{m}$ at the $\text{Co}_2\text{MnGa}/\text{Bi}_{0.2}\text{Sb}_{1.8}\text{Te}_3$ interfaces. Considering the fact that the thickness of the interdiffusion layer is much smaller than those of the Co_2MnGa layers ($\sim 0.7 \text{ mm}$), the atomic diffusion hardly influences the transverse thermoelectric property of Co_2MnGa in macroscale. Similar results were obtained in the $\text{Co}_2\text{MnGa}/\text{Bi}_2\text{Te}_3$ multilayers as shown in Supplementary Fig. 1d.

In the revised manuscript, to emphasize the sharpness of the boundaries of ATMLs, we have added these results in Supplementary Information and revised sentences of the main text on Lines 111-114. We have also added an author (Hossein Sepehri-Amin) in light of the contribution to these experiments.

(Comment 2-4)

The data of specific heat using the differential scanning calorimetry should be provided.

(Answer 2-4)

In accordance with your comment, we have added the measured values of the specific heat, thermal diffusivity, and density for estimating the thermal conductivity in Supplementary Table 1.

(Comment 2-5)

In Fig. 3 and Fig. 4, the direction of infrared camera imaging and the configuration of testing should be clearly marked in the diagram. Moreover, it is preferable to label the material composition and temperature changes of each region in the thermal imaging image one-to-one in the image (Fig. 3c,d,k,i).

(Answer 2-5)

Following your suggestion, we have revised Figs. 3 and 4 in the main text and Supplementary Figs. 4 and 5.

(Comment 2-6)

What are the temperature and spatial resolutions of the LIT technique?

(Answer 2-6)

Generally, the temperature resolution of the lock-in thermography (LIT) is less than 0.1 mK. The spatial resolution depends on the magnification of lens of an infrared camera and is generally in the range of 5-20 μm . In this study, the spatial resolution is around 20 μm to measure thermal images for the large samples.

In accordance with your comment, we have added the description of temperature and spatial resolutions of LIT (Line 129) and viewing area size of our LIT system (Lines 383-384) in the revised manuscript.

(Comment 2-7)

Some minor issues are worth considering for improving this manuscript's readability.

(Answer 2-7)

We really appreciate your careful review to improve the quality of our manuscript. Referring to Comments 2-7-1 to 2-7-6, we have carefully revised the manuscript.

(Comment 2-7-1)

The manuscript focuses on the cooling phenomena of thermoelectric materials (whether longitudinal or transverse), should the title highlight the cooling-related effects more prominently?

(Answer 2-7-1)

As you pointed out, the visualization of transverse thermoelectric cooling hybridized by AEE is one of the main data. However, we consider that another important novelty of this study is the quantitative evaluation of magneto-thermoelectric modulation of dimensionless figure of merit for the transverse thermoelectric conversion, which was quantified only by measuring the transverse thermopower due to ANE.

Thus, we would like to keep the title as it is. To emphasize the focus of our study, we have revised the sentences on Lines 288-295 and added the simulation of dimensionless figure of merit for the hybrid transverse magneto-thermoelectric generation in Supplementary Fig. 6.

(Comment 2-7-2)

On lines 28 and 29, “...a thermoelectric module based on the Seebeck/Peltier effect usually consists of a bunch of thermocouples, made of p-type and n-type...”, where the “thermocouples” used here seems to be misplaced.

(Answer 2-7-2)

Thank you very much for kindly correcting our description. In the revised manuscript, we have revised the sentences on Lines 29-30 following your suggestion.

(Comment 2-7-3)

On line 36, “Separating the J_c and J_q directions not only reduces the number of electrodes but also enhances the...”. For both Seebeck effect-based and Nernst effect-based devices, when the number of thermoelectric legs in the device is the same, the number of electrodes seems to be the same.

(Answer 2-7-3)

In the transverse thermoelectric conversion, including the Nernst effects, owing to the orthogonal relationship between a charge and heat current, the electromotive force is increased by simply extending the length of a single thermoelectric leg along a direction perpendicular to the heat current. Thus, a single block, sheet, and wire work as a thermoelectric module based on the transverse thermoelectric conversion without fabricating a thermopile structure, which enables to greatly reduce the number of electrode contacts; in principle, only two electrodes at each end of the thermoelectric leg are required at the minimum, as shown in ref. 4 in the revised manuscript [Uchida, K. & Heremans, J. P., *Joule* **6**, 2240 (2022)].

To make it easier for readers, we have revised the sentences on Lines 36-38 with adding the reference.

(Comment 2-7-4)

On lines 60 and 61, “a large transverse conversion by ODSE/ODPE has been realized using conventional thermoelectric materials.”, where relevant references may be needed.

(Answer 2-7-4)

Following your comment, we have added the several references on Line 63 of the revised manuscript.

(Comment 2-7-5)

On line 112, The simulation details related to θ and R may need to be provided.

(Answer 2-7-5)

We have added the details of the analytical simulation of transport properties for transverse thermoelectric conversion in ATMLs in Supplementary Note 1 and cited it on Line 101 of the revised manuscript.

(Comment 2-7-6)

On line 116, “...well as polycrystalline Co_2MnGa [39,41].”, Co_2MnGa in reference [39] is reported as a

film while in reference [41] it is polycrystalline bulk, the latter could be more comparable to the currently studied Co_2MnGa as they are all fabricated using the SPS method.

(Answer 2-7-6)

In ref. 40 in the revised manuscript (ref. 39 in the original manuscript) [Zhou, W. et al. *Appl. Phys. Lett.* **122**, 062402 (2023)], the authors use bulk polycrystalline Co_2MnGa slabs prepared using the spark plasma sintering method [see Fig. 3(a),(b),(d) of this paper]. Thus, we believe that both references are suitable for comparison of transport properties with our Co_2MnGa slab.

[Reply to the comments of Reviewer #3]

We are grateful to Reviewer #3 for their comments, which were very helpful to us in improving our manuscript. Our responses to your individual comments are provided below.

(Comment 3)

This study investigates the anomalous Nernst effect (ANE) and the off-diagonal Seebeck effect (ODSE) in artificially tilted multilayers (ATMLs). Transverse thermoelectric effects such as ANE have attract significant attention due to their potential to reduce device dimensions in thermoelectric generators. The authors employ a large ANE in topological materials and ODSE in ATMLs cooperatively to enhance transverse thermoelectric performance. This approach is novel. They present high-quality data on charge-heat current conversion and vice versa and discuss an intriguing hybrid action between ODSE and ANE in ATMLs. The manuscript is well-written, and the scientific discussion is robust. However, its impact appears limited for general readers due to the following reasons:

1. Modulation of the transverse thermoelectric effect by magnetic fields is small as seen in Fig. 6c.
2. The experimental design and results in Figs. 1-4 closely resemble those in their recent work (K. Uchida et al., *Adv. Energy Mater.* 2024, 14, 2302375), where they had reported high thermoelectric performance in ATMLs.

In conclusion, while the perspective, data, and discussion quality are excellent, the results lack significant impact and do not meet the standards expected by Nature Communications. This manuscript might find better reception in journals specializing in applied physics or material science, where it could benefit other researchers.

(Answer 3)

Thank you very much for giving us the insightful comments to confirm the solidity of this work. In accordance with your comments, we have deeply considered and revised the manuscript. The authors' reply to the comments and revised parts of the manuscript are detailed below.

(Comment 3-1)

Modulation of the transverse thermoelectric effect by magnetic fields is small as seen in Fig. 6c.

(Answer 3-1)

We agree with your opinion that the magnetic-field-induced modulation of transverse thermoelectric conversion performance in this study is still small. However, we would emphasize that our proposal of hybrid action of the anomalous Nernst effect (ANE) and off-diagonal Seebeck effect (ODSE) reveals the potential for further improving the modulation of the transverse thermoelectric performance. To further clarify this, we have simulated the dimensionless figure of merit for hybrid transverse thermoelectric conversion in artificially tilted multilayers (ATMLs) $z_T T$ with changing the transverse thermopowers of ODSE and ANE when other transport parameters are fixed to those of $\text{Co}_2\text{MnGa}/\text{Bi}_{0.2}\text{Sb}_{1.8}\text{Te}_3$ ATMLs. Supplementary Fig. 6a in the revised manuscript shows the dependence of the absolute value of the effective anomalous Nernst coefficient in ATML, $|\tilde{S}_N|$, on the magneto-thermoelectric modulation of $z_T T$, i.e., $\Delta(z_T T) = \tilde{S}_N^2 + 2|S_{\text{OD}}\tilde{S}_N|$, at various magnitudes of the transverse thermopower due to ODSE $|S_{\text{OD}}|$. While a dramatic improvement in $|\tilde{S}_N|$ from the current record-high value (e.g., $>20 \mu\text{V}/\text{K}$) would only yield the $\Delta(z_T T)$ value of 0.01 without ODSE, the presence of ODSE not only significantly improves the $\Delta(z_T T)$ value but also further increases the contribution of ANE on $\Delta(z_T T)$, i.e., the slope in Supplementary Fig. 6a, resulting in $\Delta(z_T T) \sim 0.05$ (~ 0.13) at $|\tilde{S}_N| = 7 \mu\text{V}/\text{K}$ ($20 \mu\text{V}/\text{K}$) and $|S_{\text{OD}}| = 100 \mu\text{V}/\text{K}$ at room temperature. Although the $|S_{\text{OD}}|$ value measured in this study is

only about 50 $\mu\text{V}/\text{K}$ due to the unoptimized Seebeck coefficient of magnetic materials, over 100 $\mu\text{V}/\text{K}$ of $|S_{\text{OD}}|$ has been reported in many ATMLs in previous studies. Thus, further improvement of $|S_{\text{OD}}|$ in magnetic-material-based ATMLs should be practical.

We believe that the above discussion has great impact on both spin caloritronics and thermoelectrics communities. Our work motivates an interdisciplinary effort of research on ANE and on magnetic materials with the large Seebeck coefficient for transverse thermoelectric applications, where the latter has received little attention so far.

To clearly describe the importance of this work, in the revised manuscript, we have added above discussion on Lines 288-293.

(Comment 3-2)

The experimental design and results in Figs. 1-4 closely resemble those in their recent work (K. Uchida et al., *Adv. Energy Mater.* 2024, 14, 2302375), where they had reported high thermoelectric performance in ATMLs.

(Answer 3-2)

Indeed, the experimental design in this study resembles that in ref. 37 in the revised manuscript [Uchida K. et al., *Adv. Energy Mater.* 14, 2302375 (2024)] due to the same symmetry of interconversion of heat and charge currents in the ordinary Nernst effect (ONE) and ANE. However, this study has clear novelty as described below.

Importantly, the mechanism and functionalities obtained from ONE and ANE are different. ONE has been theoretically and experimentally shown to scale with carrier mobility [Behnia, K. and Aubin H., *Rep. Prog. Phys.* 79, 046502 (2016)]. On the other hand, in the case of ANE, such simple scaling does not exist since its mechanism is related to the Berry curvature of Bloch function, whose complex nature complicates materials exploration for ANE, conversely, gives extra room for development. In addition, ANE has an advantage of operating at much smaller magnetic field values than that for ONE or in the absence of a magnetic field when using magnetic materials with finite coercivity and remanent magnetization as mentioned on Lines 45-49 and 320-324 in revised manuscript.

With the development of the topological materials science, materials exhibiting large ANE have been discovered one after another, and this has become a major trend in the solid-state physics. However, ANE itself is still far from practical application; the order of 0.1 of $z_{\text{T}}T$ cannot be achieved without an unrealistically large anomalous Nernst coefficient of 70-80 $\mu\text{V}/\text{K}$ when using only ANE. The experiments and discussions shown in this study provides a new opportunity to link active basic research on ANE with device application; the order of 0.1 of $z_{\text{T}}T$ can be realized by hybrid action of ANE and ODSE with realistic thermopower of ANE and ODSE (see Supplementary Fig. 6b).

Thus, even if the experimental design is similar, we believe that there are significant research developments in both basic and applied research aspects that can be gained from the superior features of ANE and its potential in ATMLs.

In the revised manuscript, we have added above discussion on Lines 49-51, 67-69, and 293-297.

(Comment 3-3)

Minor comment: The sentence in Line 132, "During the LIT measurements, an in-plane H with the magnitude of H was applied along the short axis of the sample (y axis in Fig. 3a,i) using an electromagnet," should be relocated since no H field was applied in Fig. 3.

(Answer 3-3)

We appreciate your careful review. Following your comment, we have moved the description on the magnetic field application to Lines 174-175 in the revised manuscript.

[List of corrections]

Author list

We have added one contributing author who performed additional microstructure characterization of the $\text{Co}_2\text{MnGa/Bi}_{10.2}\text{Sb}_{1.8}\text{Te}_3$ and $\text{Co}_2\text{MnGa/Bi}_2\text{Te}_3$ stacks as following.

(Original manuscript)

Takamasa Hirai, Fuyuki Ando, and Ken-ichi Uchida

(Revised manuscript)

Takamasa Hirai, Fuyuki Ando, Sepehri-Amin Hossein, and Ken-ichi Uchida

Main text

Lines 29-30.

(Original manuscript)

... usually consists of a bunch of thermocouples, made of p -type and n -type conductors, connected electrically ...

(Revised manuscript)

... usually consists of a bunch of p -type and n -type conductors connected electrically ...

Lines 36-38.

(Original manuscript)

Separating the \mathbf{J}_c and \mathbf{J}_q directions not only reduces the number electrodes but also enhances the thermoelectric output simply by upscaling the size of the materials.

(Revised manuscript)

Separating the \mathbf{J}_c and \mathbf{J}_q directions enables the enhancement of thermoelectric output simply by upscaling the size of a single thermoelectric material, which can reduce the number of electrodes and junctions in modules⁴.

Lines 49-50.

(Added sentences)

While ONE has been shown to scale with carrier mobility¹⁵, such simple scaling rules do not exist for ANE, and the material exploration for ANE is underway from various perspectives.

Line 63.

(Original manuscript)

... conventional thermoelectric materials.

(Revised manuscript)

... conventional thermoelectric materials²⁷⁻³⁶.

Lines 65-69.

(Original manuscript)

Although the hybrid transverse magneto-thermoelectric conversion based on ONE/OEE and ODSE/ODPE in ATMLs has recently been reported³⁶, hybridizing ANE/AEE has not ...

(Revised manuscript)

Although the hybrid transverse magneto-thermoelectric conversion based on ONE/OEE and ODSE/ODPE in ATMLs comprising nonmagnetic thermoelectric materials, i.e., Bi-Sb and Bi-Sb-Te, has recently been reported³⁷, hybridizing ANE/AEE and its feature superior to ONE/OEE has not been ...

Line 101.

(Original manuscript)

..., according to the equations in ref. 36.

(Revised manuscript)

..., according to the equations in ref. 36 (see Supplementary Note 1).

Lines 111-114.

(Added sentences)

Here, we confirmed that the sharp repeating boundaries in the $\text{Co}_2\text{MnGa}/\text{Bi}_{0.2}\text{Sb}_{1.8}\text{Te}_3$ stack through structural characterizations; the composition of the bulk regions of each layer was uniform and the atomic interdiffusion was limited to a region of $\sim 10 \mu\text{m}$ at the $\text{Co}_2\text{MnGa}/\text{Bi}_{0.2}\text{Sb}_{1.8}\text{Te}_3$ interfaces (Supplementary Fig. 1).

Lines 122-123.

(Added sentences)

The σ (κ) value of our polycrystalline Co_2MnGa is almost same as (smaller than) that of single-crystalline Co_2MnGa ³⁸.

Lines 128-129.

(Original manuscript)

... by an external periodic input with high temperature and spatial resolutions⁴².

(Revised manuscript)

... by an external periodic input with high temperature ($< 0.1 \text{ mK}$) and spatial (around $10\text{-}20 \mu\text{m}$) resolutions⁴³.

Line 138.

(Deleted sentences)

During the LIT measurements, an in-plane \mathbf{H} with the magnitude of H was applied along the short axis of the sample (y axis in Fig. 3a,i) using an electromagnet.

Lines 174-175.

(Original manuscript)

... the LIT images captured at non-zero H .

(Revised manuscript).

... the LIT images captured under an application of in-plane \mathbf{H} with the magnitude of H along the short axis of the sample (y axis in Fig. 3a,i) using an electromagnet.

Lines 288-295.

(Added sentences)

Supplementary Fig. 6a shows the dependence of the absolute value of \tilde{S}_N on the magneto-thermoelectric modulation of $z_T T$, i.e., $\Delta(z_T T) = \tilde{S}_N^2 + 2|S_{OD}\tilde{S}_N|$, at various magnitudes of S_{OD} . The presence of ODSE not only significantly improves the $\Delta(z_T T)$ value without changing $|\tilde{S}_N|$ but also further increases the contribution of ANE on $\Delta(z_T T)$, i.e., the slope in Supplementary Fig. 6a, resulting in $\Delta(z_T T) \sim 0.05$ (~ 0.13) at $|\tilde{S}_N| = 7 \mu\text{V/K}$ ($20 \mu\text{V/K}$) and $|S_{OD}| = 100 \mu\text{V/K}$ at room temperature (note that over $100 \mu\text{V/K}$ of $|S_{OD}|$ has been reported in many ATMLs in previous studies^{32-35,37}). Furthermore, as shown in Supplementary Fig. 6b, improving both \tilde{S}_N and S_{OD} can enhance the $z_T T$ value, e.g., $z_T T \sim 0.35$ at $|\tilde{S}_N| = 7 \mu\text{V/K}$ and $|S_{OD}| = 100 \mu\text{V/K}$ at room temperature.

Lines 296-297.

(Added sentences)

Especially, the exploration of magnetic materials with large S_s has received little attention so far, leading to new research on magnetic materials.

Lines 297-299.

(Added sentences)

Interface engineering in ATMLs is also promising since multilayering could increase S_N itself intentionally via an interfacial spin-orbit interaction⁶⁰.

Line 335.

(Original manuscript)

... ingots with a diameter of 20 mm were prepared ...

(Revised manuscript).

... ingots with a diameter of 20 mm and a height of 15 mm were prepared ...

Lines 338-340.

(Original manuscript)

The composition ratio of sintered Bi-Sb-Te slabs was confirmed to be $\text{Bi}_{0.2}\text{Sb}_{1.8}\text{Te}_3$ using scanning electron microscopy with energy-dispersive X-ray spectroscopy and the Co_2MnGa , $\text{Bi}_{0.2}\text{Sb}_{1.8}\text{Te}_3$, and Bi_2Te_3 alloys prepared by the above procedures are polycrystalline, suggesting their isotropic transport properties.

(Revised manuscript).

The composition ratio of sintered Bi-Sb-Te slabs was confirmed to be $\text{Bi}_{0.2}\text{Sb}_{1.8}\text{Te}_3$ using scanning electron microscopy (SEM) with energy-dispersive X-ray spectroscopy (EDS) (Cross-Beam 1540ESB, Carl Zeiss AG). The polycrystalline nature of Co_2MnGa , $\text{Bi}_{0.2}\text{Sb}_{1.8}\text{Te}_3$, and Bi_2Te_3 alloys prepared by the above procedures were also obtained by SEM-EDS.

Lines 342-343.

(Added sentences)

Here, for the $\text{Bi}_{0.2}\text{Sb}_{1.8}\text{Te}_3$ and Bi_2Te_3 slabs, two rectangular slabs with different cutout directions were prepared (see Supplementary Note 2 and Fig. 7).

Lines 380-384.

(Added sentences)

The minimum f value during the LIT measurements was determined by considering the thermal boundary conditions; we chose 0.2 Hz as a minimum f because we have confirmed that there is no influence of parasitic signals, such as the Peltier effect at the sample edges due to wiring, to the signal on the sample in the viewing area of LIT at 0.2 Hz. The viewing area of all the lock-in images is 512×260 pixels = 7.68×3.90 mm².

Lines 524-525.

(Added sentence in Author contributions)

H.S.A conducted the microstructural characterization.

- **Some minor typos in the main text have been corrected.**

Figures

Figure 2.

- We have replaced the calculated data with that for newly sintered Bi_{0.2}Sb_{1.8}Te₃ slab.

Figure 3.

- We have added the image of infrared camera in (a) and (i) and showed the region of each element of ATMLs in (c) and (k).

Figure 4.

- We have showed the region of each element of ATMLs in (a).

References

We have added the following references and renumbered:

15. Behnia, K. & Aubin, H. Nernst effect in metals and superconductors: A review of concepts and experiments. *Reports Prog. Phys.* **79**, 046502 (2016).

55. Uchida, K. *et al.* Enhancement of anomalous Nernst effects in metallic multilayers free from proximity-induced magnetism. *Phys. Rev. B* **92**, 094414 (2015).

Supplementary Information

- **We have newly added Supplementary Notes 1 (simulation of transverse thermopower in ATMLs) and 2 (investigation of the anisotropy of Bi_{0.2}Sb_{1.8}Te₃ and Bi₂Te₃ slabs made by SPS).**

Note 1 | Calculation of transport properties in artificially tilted multilayers.

Referring to ref. 1 and 2 in Supplementary Information, the thermoelectric, electrical, thermal transport parameters of magnetic/thermoelectric multilayers in the direction parallel ($S_{||}$, $\sigma_{||}$, and $\kappa_{||}$) and

perpendicular (S_{\perp} , σ_{\perp} , and κ_{\perp}) to the stacking plane are formulated using the following Supplementary Equations (1-3) on the assumption that the interfacial electrical and thermal resistances at boundaries between magnetic and thermoelectric layers are negligibly small:

$$S_{\parallel} = \frac{R\sigma_M S_{S,M} + (1-R)\sigma_{TE} S_{S,TE}}{R\sigma_M + (1-R)\sigma_{TE}}, \quad S_{\perp} = \frac{R\kappa_{TE} S_{S,M} + (1-R)\kappa_M S_{S,TE}}{R\kappa_{TE} + (1-R)\kappa_M} \quad (1)$$

$$\sigma_{\parallel} = R\sigma_M + (1-R)\sigma_{TE}, \quad \sigma_{\perp} = \frac{\sigma_M \sigma_{TE}}{(1-R)\sigma_M + R\sigma_{TE}} \quad (2)$$

$$\kappa_{\parallel} = R\kappa_M + (1-R)\kappa_{TE}, \quad \kappa_{\perp} = \frac{\kappa_M \kappa_{TE}}{(1-R)\kappa_M + R\kappa_{TE}} \quad (3)$$

Here, $S_{S,M}$ ($S_{S,TE}$) is the Seebeck coefficient, σ_M (σ_{TE}) the electrical conductivity, and κ_M (κ_{TE}) the thermal conductivity of the magnetic (thermoelectric) component and $R [= t_M/(t_M+t_{TE})]$ is the thickness ratio with t_M (t_{TE}) being the thickness of the magnetic (thermoelectric) component. When the magnetic/thermoelectric stacks are rotated with the tilt angle θ to form artificially tilted multilayers (ATMLs), the transverse thermopower due to the off-diagonal Seebeck effect (ODSE), S_{OD} , and the electrical and thermal conductivities orthogonal to each other (σ_{xx} and κ_{yy}) are respectively expressed as

$$S_{OD} = (S_{\perp} - S_{\parallel}) \sin \theta \cos \theta \quad (4)$$

$$\sigma_{xx} = \frac{\sigma_{\parallel} \sigma_{\perp}}{\sigma_{\parallel} \sin^2 \theta + \sigma_{\perp} \cos^2 \theta} \quad (5)$$

$$\kappa_{yy} = \kappa_{\parallel} \sin^2 \theta + \kappa_{\perp} \cos^2 \theta \quad (6)$$

With these parameters, the dimensionless figure of merit for ODSE $z_{OD}T$ with T being the absolute temperature is defined as

$$z_{OD}T = \frac{S_{OD}^2 \sigma_{xx}}{\kappa_{yy}} T \quad (7)$$

Note 2 | Anisotropy of electric and thermoelectric transport properties in sintered $\text{Bi}_{0.2}\text{Sb}_{1.8}\text{Te}_3$ and Bi_2Te_3 .

From the sintered cylindrical Bi_2Te_3 and $\text{Bi}_{0.2}\text{Sb}_{1.8}\text{Te}_3$ ingots with a diameter of 20 mm and a height of 15 mm, two rectangular slabs, named ‘‘Slab 1’’ and ‘‘Slab 2’’, with a size of $\sim 12 \times 3 \times 1 \text{ mm}^3$ were cut out using the diamond wire saw, where the longitudinal direction of Slab1 (Slab2) was along the diameter (height) direction of the ingot, i.e., perpendicular (parallel) to the pressing direction during spark plasma sintering (SPS) (Supplementary Fig. 7a). Supplementary Fig. 7b shows the Seebeck coefficient S_s and electrical conductivity σ of Slabs 1 and 2 for Bi_2Te_3 and $\text{Bi}_{0.2}\text{Sb}_{1.8}\text{Te}_3$. In both the Bi_2Te_3 and $\text{Bi}_{0.2}\text{Sb}_{1.8}\text{Te}_3$ slabs, there was little difference in S_s between Slabs 1 and 2, while the magnitude of σ of Slab 2 was clearly smaller than that of Slab 1 even though our $\text{Bi}_{0.2}\text{Sb}_{1.8}\text{Te}_3$ and Bi_2Te_3 slabs are polycrystalline. The behavior of isotropic S_s and anisotropic σ in Bi-Sb-Te and Bi-Te prepared by SPS has been also reported in the previous study; it is not due to their intrinsic band structure but due to the particle shape of the matrix powder³. However, the observed anisotropy of σ has little impact on the transport properties of our ATMLs; the value of dimensionless figure of merit for the off-diagonal Seebeck effect calculated with the parameters for Slab 1 was almost the same as that for Slab 2 (Supplementary Fig. 7c,d and Supplementary Note 1 for calculation details).

- We have newly added Supplementary Figs. 1 (structural and chemical characterization), 6 (numerical simulation of dimensionless figure of merit for hybrid transverse magneto-

thermoelectric generation), and 7 (investigation of the anisotropy of $\text{Bi}_{0.2}\text{Sb}_{1.8}\text{Te}_3$ and Bi_2Te_3 slabs made by SPS).

Supplementary Fig. 1 | Structural and chemical characterization in Co_2MnGa -based multilayers. **a,b** Cross-sectional scanning electron microscopy with energy-dispersive X-ray spectroscopy mapping images for the $\text{Co}_2\text{MnGa}/\text{Bi}_{0.2}\text{Sb}_{1.8}\text{Te}_3$ multilayer with low (**a**) and high (**b**) magnification. Line profiles of the atomic composition across the stacking direction in the mapping images are also shown. **c,d** Results for the $\text{Co}_2\text{MnGa}/\text{Bi}_2\text{Te}_3$ multilayer.

Supplementary Fig. 6 | Simulation of dimensionless figure of merit for transverse thermoelectric generation with hybridized ODSE and ANE in ATML. **a** Dependence of the absolute value of \tilde{S}_N on the magneto-thermoelectric modulation of the dimensionless figure of merit for hybrid transverse magneto-thermoelectric conversion $z_T T$, i.e., $\Delta(z_T T) = \tilde{S}_N^2 + 2|S_{OD}\tilde{S}_N|$, at various $|S_{OD}|$ values, where \tilde{S}_N is the effective anomalous Nernst coefficient in ATML [see Equation (4) in the main text]. **b** Contour maps of $z_T T$ as functions of $|\tilde{S}_N|$ and $|S_{OD}|$. Here, $\sigma_{xx} = 3.3 \times 10^5 \text{ S/m}$ and $\kappa_{yy} = 3.7 \text{ W/mK}$ obtained for $\text{Co}_2\text{MnGa}/\text{Bi}_{0.2}\text{Sb}_{1.8}\text{Te}_3$ ATMLs were used for estimating $\Delta(z_T T)$ and $z_T T$.

Supplementary Fig. 7 | Dependence of electrical and thermoelectric transport properties of sintered $\text{Bi}_{0.2}\text{Sb}_{1.8}\text{Te}_3$ and Bi_2Te_3 on pressing direction. **a** Schematic of the preparation of $\text{Bi}_{0.2}\text{Sb}_{1.8}\text{Te}_3$ and Bi_2Te_3 slabs. **b** S_s and σ of Slabs 1 and 2 for $\text{Bi}_{0.2}\text{Sb}_{1.8}\text{Te}_3$ and Bi_2Te_3 . **c** θ dependence of simulated $z_{OD}T$ for $\text{Co}_2\text{MnGa}/\text{Bi}_{0.2}\text{Sb}_{1.8}\text{Te}_3$ ATML at $R = 0.43$. The black arrow shows $z_{OD}T$ at θ of our $\text{Co}_2\text{MnGa}/\text{Bi}_{0.2}\text{Sb}_{1.8}\text{Te}_3$ ATML ($= 30^\circ$). The calculated value of $z_{OD}T$ is not changed by the difference in transport properties between Slabs 1 and 2. **d** $z_{OD}T$ at $R = 0.43$ and $\theta = 30^\circ$ (32°) for $\text{Co}_2\text{MnGa}/\text{Bi}_{0.2}\text{Sb}_{1.8}\text{Te}_3$ ($\text{Co}_2\text{MnGa}/\text{Bi}_2\text{Te}_3$) ATML.

- The revisions of Figures in the original manuscript are as follows.

Supplementary Table 1.

- We have added the value of thermal diffusivity, specific heat, and density.

Supplementary Figure 3.

- We have replaced the calculated data with that for newly sintered Bi_2Te_3 slabs.

Supplementary Figure 4.

- We have added the image of infrared camera in (a) and (i) and showed the region of each element of ATMLs in (c) and (k).

Supplementary Figure 5.

- We have showed the region of each element of ATMLs in (a).

- We have changed the numbering order of Figures according to the above additions.

We are grateful to all the reviewers for their constructive and helpful comments. Our responses to the reviewers' individual comments are provided below.

[Reply to the comments of Reviewer #1]

(Comment 1)

The authors have addressed all my previous concerns in a satisfactory way. I recommend the publication of the revised manuscript.

(Answer 1)

We really appreciate for your careful review and positive evaluation.

[Reply to the comments of Reviewer #2]

(Comment 2)

The authors have made satisfactory revision. It can be acceptable as is.

(Answer 2)

We really appreciate for your careful review and positive evaluation.

[Reply to the comments of Reviewer #3]

(Comment 3)

I appreciate the authors' effort. The revised manuscript clearly explains the importance of the hybridization effect, as shown in Supplementary Fig. 6. However, I still do not fully agree with the authors' response.

(Answer 3)

Thank you very much for your understanding and additional constructive comments. In accordance with your comments, we have deeply reconsidered and revised the manuscript. Our responses to your comments and the revised parts of the manuscript are provided below.

(Comment 3-1)

Regarding the first point (Modulation of the transverse thermoelectric effect by magnetic fields is small), I find the authors' rebuttal to be reasonable. I now understand that $\Delta z_{TT} \sim 0.05$ under ideal conditions is too significant to ignore, and this result cannot be achieved solely through ANE/AEE. This is likely to attract attention from the community. However, I am concerned that this approach cannot reverse the output sign through magnetization reversal, thus losing the advantage of ANE/AEE. I would appreciate further comments from the authors on this.

(Answer 3-1)

We appreciate your highly positive evaluation to the additional contents about the dimensionless figure of merit in our manuscript. As for your new concerns, we would like to offer the following thoughts. It is true that the anomalous Nernst/Ettingshausen effect (ANE/AEE) has a unique feature that the sign of its output is reversed by the magnetization reversal. However, there have been few examples of such a feature being directly utilized and suggested for thermoelectric applications. The main reason for this is that, in the application of ANE to thermoelectric generators and heat flux sensors, the device performance is determined by the magnitude of an output charge current or voltage, not by its sign. When constructing an ANE-based transverse thermoelectric module with a zigzag thermopile structure [e.g., Sakuraba, Y. et al., *Appl. Phys. Express* **6**, 033003 (2013), Wang, J. et al., *Sci. Rep.* **11**, 11228 (2021)], if the magnetization direction of each thermoelectric leg could be alternately reversed, it would be a great advantage for assembling the module, but it is difficult to achieve such a magnetization configuration by applying a magnetic field. In contrast, by alternately arranging artificially tilted multilayer (ATML) slabs with reversed tilt angles and connecting them in series, a thermopile module can be constructed using single ATML because the sign of the thermoelectric output due to the off-diagonal Seebeck/Peltier effect (ODSE/ODPE) can be reversed by the 180° reversal of the relative angle between the tilting direction and input heat/charge current (please see Ando, F. et al., <https://arxiv.org/abs/2402.18019>). The sign reversal of an output heat current due to AEE is useful for active thermal management and temperature control, but this can be realized simply by reversing the direction of an input charge current even in conventional Peltier modules as well as in ODPE and AEE. Therefore, we believe that our proposed hybrid transverse magneto-thermoelectric conversion in magnetic-material-based ATMLs is as versatile as and much more efficient than ANE/AEE. In light of your comments, we have added the sentence on Lines 61-63 in the revised manuscript to emphasize the feature of ODSE/ODPE.

(Comment 3-2)

As for the second point (The experimental design and results in Figs. 1-4 closely resemble those in their recent work), I feel the rebuttal is somewhat insufficient. To further distinguish this work from the authors' previous work, would it be possible to include a visualization of the magnetization-dependent transverse thermoelectric signal in the cross-sectional configuration?

(Answer 3-2)

Thank you very much for your constructive comments. Following your suggestion, we have performed the lock-in thermography (LIT) measurements in the cross-sectional configuration with applying a magnetic field in the vertical direction (z axis in Fig. 3a), which have never been carried out in previous studies. When the magnetization of Co_2MnGa aligns along the vertical magnetic field, the heat flow due to ODPE and AEE are hybridized in the cross-sectional configuration. Figure 4a,b in the revised manuscript represents the magnetic-field-odd-dependent component of LIT images for $\text{Co}_2\text{MnGa}/\text{Bi}_{0.2}\text{Sb}_{1.8}\text{Te}_3$ ATML. This result shows in more detail than that in the top-side configuration how the heat current generated by AEE in Co_2MnGa contributes to the total temperature change in ATML, where the temperature distribution generated by AEE is quite different from that by ODPE. Such a visualization of the temperature distribution in ATML is useful for optimizing the ATML structure for more efficient hybrid transverse magneto-thermoelectric conversion. Similar results were obtained in $\text{Co}_2\text{MnGa}/\text{Bi}_2\text{Te}_3$ ATML, as shown in Supplementary Fig. 5a,b. In the revised manuscript and Supplementary Information, we have added the above results and revised the sentences on Lines 169, 173-176, 184-195, and 384-386.

Again, thank you so much for your insightful comments improving the quality of our manuscript.

[List of corrections]

Main text

Lines 61-63.

(Added sentences)

Here, the sign of the thermoelectric output due to ODSE/ODPE can be reversed by the 180° reversal of the relative angle between the tilting direction and input heat/charge current.

Lines 169.

(Deleted sentences)

Hereafter, we focus on the results for the top-side configuration because the signals at the sample edges in the cross-section configuration are difficult to use for quantitative discussions due to the limitation of spatial resolution of LIT³⁷.

Lines 173-176.

(Original manuscript)

... under an application of in-plane \mathbf{H} with the magnitude of H along the short axis of the sample (y axis in Fig. 3a,i) using an electromagnet.

(Revised manuscript).

... under an application of \mathbf{H} with the magnitude of H . To hybridize ODPE and AEE, \mathbf{H} was applied along the vertical direction (z axis in Fig. 3a) [horizontal short direction (y axis in Fig. 3i)] of the sample in the cross-section (top-side) configuration using electromagnets.

Lines 186-187.

(Original manuscript)

..., where $A(+H)$ [$\phi(+H)$] and $A(-H)$ [$\phi(-H)$] are A (ϕ) measured in the positive ($+y$ direction) and negative ($-y$ direction) H , respectively.

(Revised manuscript).

..., where $A(+H)$ [$\phi(+H)$] and $A(-H)$ [$\phi(-H)$] are defined as A (ϕ) measured in the positive ($-z$ or $+y$ direction) and negative ($+z$ or $-y$ direction) H , respectively.

Lines 185-194.

(Added sentences)

Figure 4a,b shows the A_{odd} and ϕ_{odd} images for $\text{Co}_2\text{MnGa}/\text{Bi}_{0.2}\text{Sb}_{1.8}\text{Te}_3$ ATML at $\mu_0|H| = 0.8$ T and $f = 10.0$ Hz (top panels) and 0.2 Hz (bottom panels) in the cross-section configuration, where μ_0 represents the vacuum permeability. The distribution of the A_{odd} and ϕ_{odd} signals inside ATML is obviously different from that arising from ODPE at zero magnetic field (Fig. 3c,d), As highlighted in the images at $f = 10.0$ Hz, the A_{odd} and ϕ_{odd} signals appear in the Co_2MnGa regions. Importantly, despite their different distributions, the H -odd-dependent components contribute to the transverse thermoelectric cooling/heating with the same symmetry as ODPE (compare the thermal images at $f = 0.2$ Hz in Figs. 3d and 4b). Hereafter, we focus on the results for the top-side configuration because the signals at the sample edges in the cross-section configuration are difficult to use for quantitative discussions due to the limitation of spatial resolution of LIT³⁷.

Lines 194-195.

(Original manuscript)

Figure 4a,b shows the A_{odd} and ϕ_{odd} images for $\text{Co}_2\text{MnGa}/\text{Bi}_{0.2}\text{Sb}_{1.8}\text{Te}_3$ ATML at $\mu_0|H| = 0.8$ T and $f = 10.0$ Hz (top-panels) and 0.2 Hz (bottom panels), where μ_0 represents the vacuum permeability.

(Revised manuscript).

Figure 5a,b shows the A_{odd} and ϕ_{odd} images for $\text{Co}_2\text{MnGa}/\text{Bi}_{0.2}\text{Sb}_{1.8}\text{Te}_3$ ATML at $\mu_0|H| = 0.8$ T and $f = 10.0$ Hz and 0.2 Hz in the top-side configuration.

Lines 384-386.

(Original manuscript)

The ATML samples were securely mounted on a phenolic resin plate with low thermal conductivity to minimize the heat leakage through thermal conduction.

(Revised manuscript).

For the LIT measurements under the vertical \mathbf{H} , the ATML samples were directly mounted on the center of an electromagnet to ensure the uniform H . For the LIT measurements under the horizontal \mathbf{H} , the ATML samples were securely mounted on a phenolic resin plate with low thermal conductivity to minimize the heat leakage through thermal conduction.

Figures

Figure 3: We have added the illustration of the z axis in (a) and (i).

We have newly added Fig. 4 and changed the numbering order of Figures according to the above additions.

Supplementary Information

Figure 4: We have added the illustration of the z axis in (a) and (i).

We have newly added Supplementary Fig. 5 and changed the numbering order of Figures according to the above additions.

We are grateful to the reviewer for their positive comments. Our responses to the reviewer's individual comments are provided below.

[Reply to the comments of Reviewer #3]

(Comment 3)

I appreciate the efforts of the authors. Regarding the first point, I now have a good understanding. For the second point, I believe the additional data has made the discussion more comprehensive. While I have some reservations about the overall impact of this paper, it represents a valuable contribution to the field. Therefore, I recommend it for the editor's careful consideration.

(Answer 3)

Thank you very much for your careful review and positive evaluation.